

# PISTON and CAMP²Ex observations of the fundamental modes of aerosol vertical variability in the Northwest Tropical Pacific and Maritime Continent's Monsoon

Jeffrey S. Reid[1], Robert E. Holz[2], Chris A. Hostetler[3], Richard A. Ferrare[3], Juli I. Rubin[4], Elizabeth J. Thompson[5], Susan C. van den Heever[6], Corey G. Amiot[7], Sharon P. Burton[3], Joshua P. DiGangi[3], Glenn S. Diskin[3], Joshua H. Cossuth[8], Daniel P. Eleuterio[8], Edwin W Eloranta[2], Ralph Kuehn[2], Willem J. Marais[2], Hal B. Maring[9], Armin Sorooshian[10], Kenneth L. Thornhill[11], Charles R. Trepte[3], Jian Wang[12], Peng Xian[1], Luke D. Ziemba[3]

1. Marine Meteorology Division, US Naval Research Laboratory, Monterey, CA 93943, USA; jeffrey.s.reid20.civ@us.navy.mil; peng.xian.civ@us.navy.mil
2. Space Science and Engineering Center University of Wisconsin-Madison, Madison. WI 53715, USA; reholz@ssec.wisc.edu; eloranta@ wisc.edu; ralph.kuhen@ssec.wisc.edu; wmarias@ssec.wisc.edu; rkuhen@ssec.wisc.edu
3. NASA Langley Research Center, Hampton VA, 23681, USA; sharon.p.burton@nasa.gov; joshua.p.digangi@nasa.gov; glenn.s.diskin@nasa.gov; richard.a.ferrare@nasa.gov; chris.a.hostetler@nasa.gov; charles.r.trepte@nasa.gov; luke.ziemba@nasa.gov
4. Remote Sensing Division, US Naval Research Laboratory, Washington DC 20375; juli.i.rubin.civ@us.navy.mil
5. NOAA Physical Sciences Lab, Boulder, CO 80305, USA; elizabeth.thompson@noaa.gov
6. Department of Atmospheric Science, Colorado State University, Fort Collins CO 80523, USA; sue.vandenHeever@colostate.edu
7. NASA Postdoctoral Program, NASA Marshall Space Flight Center, Huntsville, AL 35805;corey.g.amiot@nasa.gov
8. Office of Naval Research, Code 322, Arlington, VA 22203-1995, USA; joshua.h.cossuth.civ@us.navy.mil; daniel.p.eleuterio.civ@us.navy.mil
9. NASA Headquarters (*Retired*), Washington DC, 20546-0001, USA; hal.maring@icloud.com
10. Department of Chemical and Environmental Engineering, University of Arizona, Tucson, AZ 85721; armin@arizona.edu
11. Science Systems and Applications, Inc. at NASA Langley Research Center, Hampton, Virginia, 23681, USA; kenneth.l.thornhill@gmail.com
12. Center for Aerosol Science and Engineering, Department of Energy, Environmental and Chemical Engineering, Washington University in St. Louis, St. Louis MO, 63130, USA; jian@wustl.edu

*Correspondence to*: Jeffrey S. Reid  (jeffrey.s.reid20.civ@us.navy.mil)

**Abstract.** While most large-scale smoke advection occurs within the free troposphere, Maritime Continent smoke transport is suspected to be unique in its long-range, near-surface transport. Such a pathway likely creates strong gradients and uncertainties in interpreting satellite and model data on light extinction, air pollution, and cloud condensation nuclei. This paper documents High Spectral Resolution Lidar (HSRL) data from the 2019 ONR PISTON cruise and NASA CAMP2Ex flights that revealed Maritime Continent smoke and pollution transport pathways and heterogeneity around the Marine Atmospheric Boundary Layer (MABL) over thousands of kilometers. Observations showed that 95% of integrated aerosol backscatter occurred below 2500 m altitude. The R/V *Sally Ride* observed 50th and 84th percentile aerosol backscatter altitudes at ~600 and ~1500 m





respectively, regardless of aerosol loading. Peak backscatter values occurred within or near the MABL top, diminishing as we

approached 2-3 km altitude, but with occasional plumes reaching the melting level at 4800 m. At monsoonal scales, aerosol models largely account for the observed directional wind shear that causes altitude-dependent particle transport: near-surface particles remain in the core monsoon flow around the MABL, while at lower latitudes, aerosol layers aloft advect more eastwardly. Around the MABL, however, significant cloud-scale variability exists due to fine-scale flow, halo-entrainment-detrainment, and cold pool phenomena. Backscatter enhancements beneath individual clouds, extending to the ocean surface,

likely relate to MABL-free troposphere exchange and air-sea interaction. So while aerosol transport occurs near the surface, particle extinction heterogeneity must still be considered for in situ observations and satellite retrievals.

## 1 Introduction

Every continent hosts biomass burning receptor regimes from an ever-increasing number of large scale smoke events. Long-

range transport of smoke is often facilitated by free tropospheric advection from either direct injection (e.g. super fires in Australia, North America, and boreal Asia; Kahn et al. 2008; Fromm et al. 2010; Sofiev et al. 2012; Paugam et al. 2016; Xian et al. 2022; Eck et al. 2023) or from "land plume" formation when the upper portions of a deep and polluted Planetary Boundary Layer (PBL)/lower free troposphere are acted upon by large scale vertical wind shear, often times along shorelines or mountain ranges (e.g. Sinha et al. 2004; Reid et al. 2013; Wang et al. 2013). Once in the free troposphere, or stratosphere for volcanic

or pyro convection events, aerosol plumes are impacted by synoptic meteorological features, such as frontal activity, large-scale subsidence/lifting, tropospheric folds, etc. and transported tens of thousands of kilometers before being scavenged. Plumes even frequently reach the Arctic and Antarctic regions (e.g. Hu et al. 2013; Xian et al. 2022). Although these events are easily tracked by satellite sensors and models through Aerosol Optical Thickness (AOT), there is difficult-to-observe variability in particle vertical distribution. In particular, lower-level transport and free troposphere-PBL exchange result in the

presence of anthropogenic aerosol particles in the remote Marine Atmospheric Boundary Layers (MABL). Indeed, it has long been recognized that terrestrially generated cloud condensation nuclei (CCN) either through MABL transport or entrainment from the lower free troposphere, can dominate maritime nuclei and light scattering (e.g. Clarke et al. 1996; 2003; 2014; Quinn et al. 2017; Ross et al. 2017; Reid et al. 2022).

The Maritime Continent (MC), made up of the islands of Indonesia, Malaysia, East Timor, and the Philippines, has a distinct

biomass burning and pollution transport phenomenology. Smoke from small agricultural waste burns and smoldering peat fires have lower injection heights than fires in other regions (Labonne et al. 2007; Wang et al. 2013) and long-range transport events appear to predominately be in the MABL and lower free troposphere (Campbell et al. 2013; Chew et al. 2013; Reid et al. 2013; 2023). While other major plumes in the world are ventilated into the free troposphere for long range transport, the MC's "long-range-near-surface-maritime" plumes are in high enough concentrations to be tracked by satellite for thousands of kilometers.



This transport scenario allows for the observation of particle evolution and lifetime through their interaction with an entirely different set of meteorological phenomena than what is more commonly studied in association with long range free troposphere or stratosphere aerosol transport. In particular, MC aerosol transport is often embedded within a convectively active monsoonal system with numerous cloud regimes (e.g. Reid et al. 2013; 2023; Xian et al. 2013; Ge et al. 2017; Hilario et al. 2021).

Copious anthropogenic emissions and numerous lower intensity fires are associated with shallow plume injection heights.
Orographic effects, off-island smoke transport by the land breeze, and vertical wind shear within the boreal summer's Southwest Monsoon (SWM) system have also been hypothesized to allow plumes to reach thousands of kilometers in length with the bulk of AOT being contributed by particles below 3 km in altitude (Reid et al. 2013; 2015; 2016; Wang et al. 2013; Ge et al. 2017; Hilario et al. 2021). While major burning events of the MC can occur at any time of year with a sufficient dry spell (Reid et al. 2012), boreal summertime drought conditions such as those related to El Niño can lead to the draining and
significant burning of peatlands well past the monsoon transition (e.g. Nichol 1998; Field and Shen, 2008; Reid et al. 2012, 2023; Goldstein et al. 2020). Such events result in the highest monthly average AOT values recorded by the Aerosol Robotic Network (AERONET; Eck et al. 2019; Reid et al. 2023). Similar to biomass burning plumes, MC megacity pollution can also be transported 1000 km or more (Cruz et al. 2019; Braun et al., 2020; Hilario et al. 2020; Lorenzo et al., 2023).

The particle transport from the MC, peninsular SE Asia, and southern China through the South China, Sulu, and Philippine
Seas, to their eventual annihilation in the West Pacific monsoonal trough lends itself to being a natural laboratory of aerosol lifecycle and cloud-related process studies, including entrainment, detrainment, microphysics/chemistry, and scavenging. At the same time, the nature of regional particle diversity and evolution makes the MC's monsoon system a venue to explore new remote sensing and modeling technologies. In 2019, the Office of Naval Research (ONR) Propagation of IntraSeaonal OscillatioNs (PISTON) Intensive Observation Period (IOP) on the R/V *Sally Ride* was conducted in partnership with the
NASA Cloud, Aerosol and Monsoon Processes Philippines Experiment (CAMP[2]Ex) with its P-3 and Stratton Park Engineering Company (SPEC) Lear 35 aircraft in the vicinity of the Philippines during the Southwest Monsoon (SWM) to Northeast Monsoon (NEM) transition (Reid et al. 2023). The combined resources of these two missions allowed for the first time detailed observations of MC, East Asia, and maritime aerosol transport characteristics, particle evolution, and cloud process outcomes. Included in the missions were three High Spectral Resolution Lidars (HSRL): 1) a CAMP[2]Ex-sponsored Langley Research
Center (LaRC) HSRL-2 instrument on the NASA P3 aircraft; 2) a PISTON sponsored University of Wisconsin-Madison Space Sciences and Engineering Center HSRL (UW-HSRL) deployed to the *Sally Ride* in the Philippine Sea 600 km east-northeast of Manila; and 3) A second NASA CALIPSO sponsored UW-HSRL (henceforth MO-HSRL) deployed to the Manila Observatory, Metro Manila, Philippines. Combined, CAMP[2]Ex and PISTON allowed for the observation of volumetric variability of lower troposphere aerosol transport across the region (Hilario et al. 2021; Edwards et al. 2022; Reid et al. 2023).

It was clear during the CAMP[2]Ex and PISTON deployment that moisture, aerosol, and hence light extinction can exhibit strong variability on relatively short spatial and temporal scales. As noted in Su et al. (2008), Tacket and Di Girolamo, (2009); Eck



et al. (2014), Reid et al., (2018) and the review by Marshak et al. (2021), aerosol properties can have sharply different properties around clouds in association with so-called "Halos" (Radke and Hobbs, 1991). However, given the locality and speed of key evolving cloud properties coupled with sampling limitations from aircraft and ship, it is challenging to analyze specific physical

processes. An aircraft can measure properties at a single point and time to high precision, but only a few hundred meters or seconds away the environment can be quite different. The ship can provide a very detailed bottom up-view, but waits for the atmosphere to pass over it. As the start of further investigation to understand how MABL and cloud heterogeneity are related to coupled aerosol-cloud, meteorological, and remote sensing impacts, this paper documents the nature of SWM and NEM aerosol particle layering and heterogeneity phenomena during 2019 PISTON/CAMP$^2$Ex campaigns. This review of the

different "Animals in the coupled aerosol-cloud-ocean zoo" is necessary before we can efficiently tease out the intertwined roles of individual processes that work together to create intricate patterns in light extinction in and around the MABL. This first analysis largely relies on NASA P-3 and PISTON *R/V Sally Ride* HSRL assets to identify types and ranges of scales of the maritime environment's aerosol structures in preparation for more detailed follow-on studies.

In this survey, we make use of the strengths of different instrument datasets to paint a holistic picture of aerosol variability in

the Northwest Tropical Pacific's (NWTP) monsoon environment. We begin with an overview of available assets (Section 2), followed by a brief large-scale survey of aerosol vertical profiles from models and in situ measurements from the P-3 (Section 3). After this overview, we provide example environments measured by the airborne LaRC HSRL-2 (Section 4), limited to the analysis of three core environments (NWTP marine, MC biomass burning, and Asian pollution). The lidar observations show significant complexity in aerosol features. We use these examples to lay the groundwork for a time-series analysis from the

*R/V Sally Ride* (Section 5) with further examples highlighting specific lidar-diagnosed aerosol features. Finally, we summarize findings and discuss implications in our conclusions (Section 6).

## 2 Mission Overview, Observations, and this Investigation Design

A full description of the CAMP$^2$Ex field mission and its relationship to PISTON is provided by Reid et al. (2023). A brief summary on relevant topics drawn from this paper is provided here. The NASA CAMP$^2$Ex experiment was conducted out of

Clark International Airport, Philippines with a) 19 flights between August 25-October 5, 2019 of the NASA P-3 as the primary platform for remote sensing, state, and composition hosting instruments including the HSRL-2 (Hair et al. 2008, Burton et al. 2018); and b) 11 SPEC Lear 35 flights from September 7-29, 2019 for sampling deep convection (not used here). P-3 research flight days as well as their back-trajectory derived airmass sources were published by Hilario et al. (2021); a synopsis of the MABL and Lower Free Troposphere (LFT) airmass origins are listed in Table 1. Within the CAMP$^2$Ex study period, PISTON

was conducted with the *R/V Sally Ride* on station ~ 600 km northeast of Manila September 6-25, 2019, transiting to and from Taiwan. Of interest to this study regarding *Sally Ride* data, was the zenith-pointing and scanning University of Wisconsin Space Science and Engineering Center (SSEC) HSRL (Eloranta 2005; Reid et al. 2017), SEAPOL C-band dual-polarization



Doppler radar (Rutledge et al. 2019), and the NOAA PSL air-sea flux and near-surface meteorological and ocean time series (Fairall et al. 1996, 1997, 2003, Edson et al 2013, Sobel et al. 2021). Finally, the NASA Cloud-Aerosol Lidar and Infrared

Pathfinder Satellite Observations (CALIPSO) mission funded the long-term deployment of another, zenith pointing, SSEC HSRL at the Manila Observatory between June 2019-September 2020. Discussion of this urban environment deployment is forthcoming in works.

**Table 1.  P3 flight summary including the local day for the flight (GMT is +8 hrs). Properties are taken from surface**

**to >5 km profiles over ocean. Included is a)  Marine Atmospheric Boundary Layer (MABL; 100-400 m) and Lower free troposphere (LFT 1200-1600m).  Approximate source regions as extracted from Hilario et al., (2021): BB-Biomass burning from Borneo and Sumatra; EA-East Asia. MC-Maritime Continent; WP-Western Pacific. MX-Mixture/Other. b) Carbon monoxide values for the MABL and 1.4 km representing the lower free troposphere measured from Picarro Cavity Ring Down Spectrometer c) Volume concentration of fine and coarse mode particles measured from the Laser**

**Aerosol Spectrometer (LAS;0.1<$d_P$<1 μm and 1<$d_P$<3 μm). (d) Estimated sub-micron dry light extinction as the sum of TSI 3λ nephelometer and Radiance Research Particle Soot/Absorption Photometer (PSAP ).**

| RF | Local Day | MABL Source | Lower Free Trop Source | MABL:LFT CO (ppb) | MABL:LFT Fine LAS Vol 0.1<$d_p$<1 μm (μm³ cm³) | MABL:LFT Coarse LAS Vol 1<$d_p$<3 μm (μm³ cm³) | MABL:LFT Dry Sub micron Light Extinction (532 nm, Mm⁻¹) |
|---|---|---|---|---|---|---|---|
| 1 | 8/25/2019* | MC | WP | 108:86 | 2.5:0.9 | 2.3:1.0 | 31:9 |
| 2 | 8/27/2019 | MX | MX | 91:85 | 2.0:0.7 | 2.1:0.6 | 22:9 |
| 3 | 8/30/2019* | MC | MC | 110:96 | 3.0:1.1 | 1.7:0.6 | 31:31 |
| 4 | 8/31/2019* | MC | MC | 102:88 | 2.5:1.7 | 1.7:1.7 | 28:21 |
| 5 | 9/4/2019 | MC | MC | 121:104 | 5.5:3.4 | 2.2:2.2 | 54:31 |
| 6 | 9/7/2019* | BB | BB | 242:133 | 13.0:4.6 | 4.8:2.1 | 166:44 |
| 7# | 9/9/2019* | BB | WP | 175:112 | 7.5:1.4 | 2.9:0.8 | 83:13 |
| 8 | 9/14/2019* | WP | MX | 74:79 | 3.5:3.2 | 1.0:0.8 | 39:32 |
| 9 | 9/16/2019* | BB | BB | 678:374 | 44.0:23 | 15.3:8.0 | 698:351 |
| 10 | 9/17/2019* | BB | WP | 312:194 | 17.0:6.7 | 6.8:2.7 | 224:74 |
| 11 | 9/20/2019* | EA | EA | 138:139 | 4.5:2.6 | 3.9:1.8 | 54:20 |
| 12 | 9/22/2019* | EA | EA | 181:153 | 7.0:3.9 | 3.9:2.2 | 82:38 |
| 13# | 9/24/2019 | EA | EA | 166:154 | 5.0:3.1 | 2.9:2.1 | 54:32 |
| 14# | 9/25/2019* | EA | EA | 149:143 | 4.0:2.2 | 2.5:1.2 | 36:21 |
| 15 | 9/28/2019 | WP | WP | 77:76 | 0.4:0.1 | 0.7:0.1 | 5:3 |
| 16 | 9/29/2019 | EA | WP | 80:73 | 1.5:0.3 | 1.5:0.4 | 18:3 |
| 17 | 10/2/2019* | EA | EA | 122:177 | 5.0:8.4 | 2.0:2.1 | 56:101 |
| 18 | 10/4/2019* | MX | MX | 161:100 | 9.0:2.6 | 1.6:0.7 | 91:24 |
| 19 | 10/5/2019 | WP | WP | 75:74 | 0.5:0.1 | 0.6:0.2 | 4:1 |

*Takeoff before 0Z on this day and hence the data filename is for the day before the local day. #Flight with R/V Sally Ride overpass.



## 2.1 Model and Satellite products

To provide context to observations we utilize satellite products generated by SSEC (outlined in Reid et al., 2023) and the Navy Aerosol Analysis and Prediction System Reanalysis (NAAPS-RA; Lynch et al. (2016) and Edwards et al. (2022)). P-3 and Lear 35 aircraft tracks as well as *Sally Ride* positions and SEAPOL C-band weather radar (Rutledge et al. 2019) plots overlaid on imagery and a variety of cloud and aerosol products used here were generated on the field mission instance of NASA
Worldview (https://geoworldview.ssec.wisc.edu/) hosted at the UW SSEC. Six-hourly NAAPS data of mass ratios, extinctions, and AOTs of smoke, sea salt, dust and combined anthropogenic/biogenic fine species (ABF), as well as associated Navy Global Environmental Model (NAVGEM; Hogan et al., 2014) meteorology are used from the NRL Global Oceans Data Assimilation Experiment site https://usgodae.org.

## 2.2 P-3 Payload

A review of the full complement of instruments on the NASA P-3 aircraft is provided in the supplemental material of Reid et al. (2023). Data used here include atmospheric state (e.g., temperature, water vapor, wind, etc.) from the P-3's Vaisala dropsondes, as well as P-3 forward camera data. Mission-wide integrated plots of particle vertical profiles from the Laser Aerosol Spectrometer (LAS; Froyd et al., 2019, Moore et al., 2021) and High-Resolution Time-of-Flight Aerosol Mass Spectrometer (HR-ToF-AMS; DeCarlo et al., 2008, Crosbie et al., 2022) in Section 3 to provide an overview of the vertical
profiles of key constituents for marine, biomass burning, and Asian pollution flights.

The focus of airborne observations in this overview paper is on the HSRL-2 (Hair et al. 2008, Burton et al. 2012; 2015; 2018; Ferrare et al. 2023) to demonstrate specific vertically-resolved aerosol phenomena. A limitation of backscatter lidars is that the measurement is fundamentally ill constrained, requiring assumptions regarding the aerosol/cloud scattering characteristics to retrieve the particulate backscatter and extinction. The HSRL technique takes advantage of the molecular Doppler
broadening due to molecular Brownian motion to separate the molecular return from particulate (aerosol/cloud) backscatter (Eloranta, 2005). The ratio of total to molecular backscatter leads to a calibrated particulate backscatter signal as a function of range, and the slope of molecular signal relates directly to the particulate extinction. Compared to other lidar types, the HSRL provides excellent daytime sensitivity due to its improved solar background rejection resulting in more tractable range dependent error propagation than more commonly used elastic backscatter or Raman systems. As noted in Reid et al. (2017)
and clearly demonstrated in this current work, while the vertical profile of extinction derived from the HSRL is a valuable product, the calibrated aerosol backscatter product from HSRL systems has greater vertical and temporal resolution, precision, and accuracy. Thus, for the topics examined here, the focus is more towards the application of aerosol backscatter products.

The LaRC HSRL is what is termed a $3\beta+2\alpha+3\delta$ system, providing calibrated aerosol backscatter and extinction ($\alpha$) at 355 and 532 nm, and attenuated backscatter ($\beta$) and depolarization ($\delta$) at 355, 532, and 1064 nm (i.e. including both aerosol and
molecular signal). The HSRL-2 acquires profiles at a rate of 200 profiles per second. Profiles are accumulated to 100-profile



averages in hardware, making the finest available temporal profile spacing of 0.5 s. To provide adequate signal-to-noise ratios, level 2 aerosol backscatter and depolarization products are further averaged over 10 s, and the native 1.25-m vertical bins are averaged to 30 m. At the P-3's flight speed of ~150 m/s, the effective horizontal resolution is 1500 m. Extinction and lidar ratio estimates are also reported at 10 seconds/30 meters but require a longer 60-second integration (7500 m on center-at flight speed) and with a 300 meter vertical window. Other derived products include estimates of planetary boundary layer (PBL) heights, which use the vertically resolved profile measurements of aerosol backscatter, and then secondarily derived aerosol mixed layer heights during the daytime (Scarino et al. 2014).

**2.3 R/V _Sally Ride_ Payload**

The ONR PISTON campaign sent the _R/V Sally Ride_ in cooperation with CAMP²Ex to study air-sea interaction and convection and their relationship to Madden Julian Oscillation's summertime manifestation, the Boreal Summer Intraseasonal Oscillation (BSISO; e.g. Jiang et al. 2004). The _R/V Sally Ride_ was thus equipped with a complement of turbulent flux, ocean sensing, and radar systems (overviewed by Sobel et al. 2021 for the 2018 cruise; all but the underway CTD data were repeated in the 2019 cruise). PISTON data are posted on the joint CAMP²Ex/PISTON data repository as https://doi.org/10.5067/SUBORBITAL/PISTON2019-ONR-NOAA/RVSALLYRIDE/DATA001 and are also accessible at https://www-air.larc.nasa.gov/missions/camp2ex/index.html; last accessed April 1, 2025). Data were continuously collected during the cruise while it was in international waters. Of particular interest here is the deployment of the UW-HSRL configured for both zenith pointing and occasional scans from the surface (0º horizontal) up to 17º in the MBL. This HSRL is very similar in nature to the one used in the NASA SEAC⁴RS project described in Reid et al. (2017) with the addition of the horizontal scanning capability, and the analysis here will follow a similar methodology. This HSRL provides aerosol backscatter at 532 nm and elastic backscatter at 1064 nm. Because the MABL, its entrainment zone, and lower free troposphere are within the region where the UW-HSRL suffers from incomplete overlap of the transmitted beam and receiver field of view, reliable lidar ratios in the area of interest are laborious to obtain. The focus here is on aerosol backscatter, and when needed, extinction estimated with the lidar ratios estimated from the HSRL-2. An ongoing study is underway on the HSRL's side scanning capabilities to develop more reliable lidar ratios in the MABL to mitigate the uncertainties of the near field (0-5 km) where the receiver is out of focus.

Mission-wide figures and statistics at times utilize a 300-second cloud screened product at 30 m vertical resolution. Considering advection for a typical 8 m/s wind speed, 30-second and 300-second averaging is equivalent to 240-m and 2400-m horizontal resolutions, respectively. This system was determined to have a precision in aerosol backscatter of better than $2*10^{-7}$ $(m\,sr)^{-1}$ and $2*10^{-8}$ $(m\,sr)^{-1}$ for these temporal averages respectively, more than an order of magnitude lower than Rayleigh backscatter ($1.16x10^{-5}$, $7.7x10^{-6}$, and $4.4x10^{-6}$ $(m\,sr)^{-1}$, at 1, 5, and 10 km). Quick-look data can be viewed and downloaded from https://hsrl.ssec.wisc.edu/. For specific cases, resolution of up to 5 seconds and 15 m are shown. In addition





to the UW- HSRL, in this paper we make use of Vaisala radiosondes released every 3 hours and mean meteorology measured by the NOAA PSL bow mast sensors.

## 2.4 Cross comparison between the P-3 and *Sally Ride* HSRLs

Because of how the UW-HSRL scan pattern was set and to maintain safety for the P-3, there were no direct overflights with contiguous data on the two platforms. The closest the P-3 and *Sally Ride* came to joint lidar observations for these cases was on the order of 3-5 km in the horizontal. A full discussion of the combined dataset is part of an ongoing analysis. However, here in the methods section it is worth briefly comparing nearby data when the P-3 flew near the *Sally Ride* to demonstrate consistency in aerosol backscatter values. The flight on September 24th observed a combination of maritime and Asian polluted

air masses. Nevertheless, the comparison of HSRL-2 data from the P-3's "top down" and UW-HSRL *Sally Ride*'s "bottom up" points of view is quite enlightening regarding the nature of lower free troposphere aerosol variability with overall good agreement of the calibrated backscatter between the instruments.

On September 24th, 2019 [Research Flight (RF) 13] the P-3 passed 4 km just to the south of the *Sally Ride*, in an area devoid of clouds (Fig. 1(a)). At the surface, winds were NE at 6 m/s, with a relative humidity (RH) of 74%, and the air-sea temperature

difference was slightly unstable at 0.5°C. These conditions are such that we do not expect MABL roll structures to form, and the near absence of clouds inhibits MABL free troposphere exchange (Lemone, 1973; Salesky et al., 2017). In every sense, this should be the most uniform of environments, and that is what is observed. In Fig. 1(b), we compare P-3 HSRL-2 in the standard 10-second increments in small markers centered on the closest position to the *Sally Ride*. For the *Sally Ride's* UW-HSRL, we chose 5-second samples around a 2-minute window when the lidar was pointed vertically. Data for the two platforms

match exceptionally well in the MABL ML at ~2.3 (Mm sr)$^{-1}$ at the surface and with the SSEC-HSRL particulate backscatter within the range of HSRL-2 values. Both HSRLs also show a typical increase in backscatter in the MABL presumed due to humidity, as well as good agreement in the height of the inversion at ~550-600 m. The strength of the backscatter at the top of the inversion varies more significantly due to strong non-linearities in hygroscopicity at high RH values (as discussed later in this paper), but the variability in both the P-3 and *Sally Ride* platforms overlap. Above the inversion is a lesser elevated layer

between 800-1400 m. Here too there is some variability, but again the individual P-3 profiles and *Sally Ride* profiles overlap.



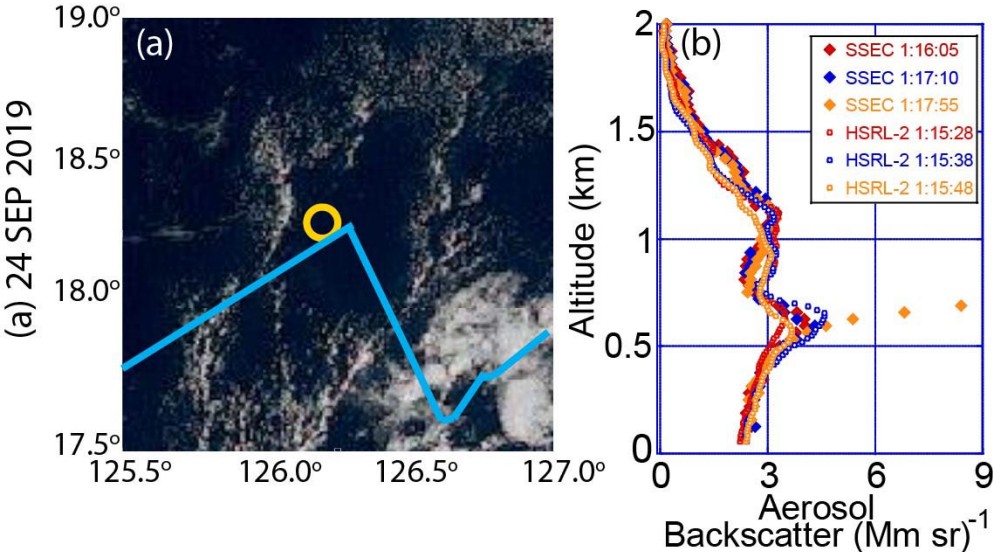

**Figure 1: Overview of when the NASA P3 flew in proximity to the Sally Ride on September 24[th] at ~ 1:15 UTC. Included is (a) the nearest RGB image from Himiwari 8, with the Sally Ride location as a yellow circle and the P3 track in blue; (b) aerosol backscatter associated with the nearest approach of the P3 including within a 2 minute window for selected samples when the SSEC HSRL is pointed vertically and within 3 nearest cloud free samples from the P3's HSRL-2.**

## 3 Results 1: Regional aerosol properties and transport

Here we provide brief context to the regional environment explored by HSRL data in the following sections. An overview of the CAMP²Ex/PISTON mission meteorology and aerosol fields is provided in Reid et al. (2023) and its associated supplemental materials. An overview of transport patterns for individual CAMP²Ex flights is provided in Hilario et al. (2021), and more regional overviews of the vertical variability in aerosol, thermodynamic and wind parameters in the region can be found in Xian et al. (2013), Bukowski et al. (2017), and Reid et al. (2012; 2023). In short, the region is dominated in the boreal summer months by the southwesterly flow of the SWM, followed by a monsoon transition to the NEM in late September to early October. For the CAMP²Ex/PISTON study season, the monsoon transition happened abruptly between September 20-22, 2019, about two weeks earlier than the climatological median. The large-scale aerosol transport pattern is elucidated in Fig. 2 where the average NAAPS-RA AOT is plotted with surface (black) and 700 hPa/~3 km (magenta) winds for SWM and NEM CAMP²Ex mission days, in (a) and (b) respectively. Also included are the P-3 flight numbers located in the primary areas of study. To map receptor locations for biomass burning emissions, corresponding wet deposition fields of the smoke species in NAAPS-RA are provided for the two monsoon periods in Fig. 2(c) and (d).

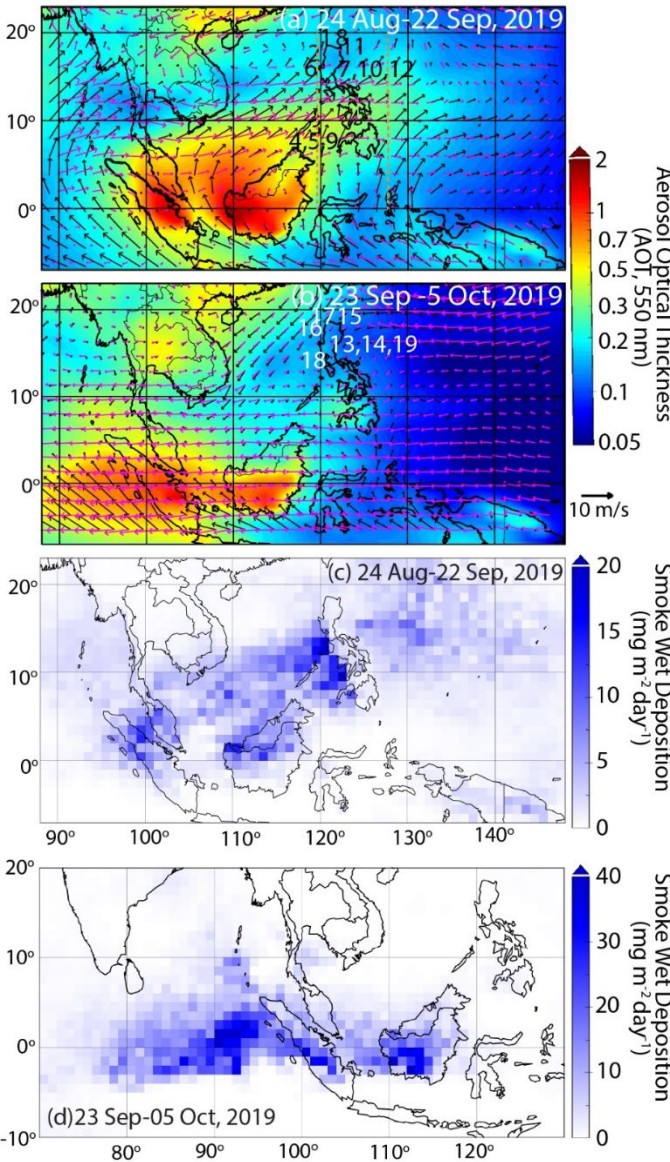

**Figure 2. Overview of the CAMP²Ex/PISTON study area for the (a) Southwest and (b) Northeast Monsoon periods of study. Included is the base average Aerosol Optical Thickness (AOT) from the NAAPS reanalysis, with (black arrows) NAVGEM surface winds; and (magenta) 3000 m AGL winds. Included are the NAAPS reanalysis smoke wet deposition flux over the study region for these two periods. Included in (a ) in dotted orange are the cross-section meridians shown in Fig. 3.**





Biomass burning and pollution emissions of the MC are dominated by Indonesia, Malaysia and Singapore. During the SWM, these emissions are carried north through the South China and Sulu Seas. Smoke wet deposition is at a maximum along the
western coast of the Philippine archipelago islands due to upstream convection induced by the islands. The Philippines then adds its own emissions to the airmass, as they are further transported and scavenged in the NWTP trough farther east. Smoke was observed from southern Japan to Guam after 4000+ km of transport. Notable in Fig. 2(a) is directional vertical wind shear around the Philippines, indicated by the difference in the surface and 700-hPa winds. Smoke near the surface is typically advected from Borneo and Sumatra towards Luzon, whereas air in the LFT has a stronger westerly component. As discussed
in Reid et al. (2013) and Bukowski et al. (2017), air above the middle troposphere overall reverses direction to northeasterly. Because of directional wind shear, back trajectories for air above the MABL are more likely to originate from Peninsular Southeast Asia than Borneo and Sumatra.

As part of the transition to the NEM, the monsoonal trough orientates from southeast to northwest in the NWTP to a more zonal orientation through Indonesia into the Indian Ocean. An opposite flow pattern to the SWM is then established (Fig. 2(b)).
Notable is that emissions from East Asia are transported along the coast to the southwest as well as some residual smoke in and around Mindanao. Biomass burning emissions from Indonesia, Malaysia and Singapore are advected into the Indian Ocean where they are scavenged in the wintertime monsoonal trough. In both monsoon periods, vertical directional wind shear influences the vertical distribution of aerosol particles in the region. As discussed in Reid et al. (2013) and shown specifically for CAMP$^2$Ex in the trajectory analysis of Hilario et al. (2021), during the SWM the MABL related air masses more typically
originate from Peninsular Southeast Asia such as from Malaysia, Thailand, Cambodia and southern Vietnam.

The impact of vertical wind shear on fine mode aerosol vertical distribution is provided in Fig. 3, that shows the 119.5° and 127.5° meridional cross section of NAAPS-RA simulated fine mode mass concentration and horizontal winds for periods examined in this paper. Rows (a) and (b) represent September 7 and 17, 2019, the two largest sampled biomass burning events, respectively; row (c) is a mixed marine case where the P-3 and *Sally Ride* operated together; row (d) is a clean marine case
north of Luzon sampled by the P-3; and row (e) is the most significant Asian pollution case sampled. These events match the general flow and AOT pattern as shown in Fig. 2(a) and (b) with the meridional transects indicated as dashed orange lines, differing only in more westward or eastward axes of the plumes. Examining the cross-sections, we see what one would expect for a sheared environment. For the two biomass burning cases ((a), (b)), smoke has a latitudinal dependence in altitude, with elevated LFT smoke around 5-12° N, followed by the MABL dominated transport north of 15°. These plume areas are clearly
aligned with areas of wind speed and directional shear in the vertical. Comparing the western to the eastern side of the Philippines, the move towards higher aerosol altitudes for lower latitudes is also visible. For the mixed (c), clean marine (d), and most polluted Asian (e) cases of Sept. 24, 27, and Oct 2, 2019, respectively, we also see consistently higher altitudes in the aerosol plumes to the south and east.







**Figure 3. Height cross-sections of NAAPS-RA fine mode aerosol mass concentrations along the 120.5 º and 127.50º meridians for cases explored in this paper. Overlaid is the NAVGEM wind vectors pointing towards the direction towards (i.e., wind up is towards N/southerly wind, towards right is East/westerly wind). Indicator lines for these meridians are shown in Figure 1(a). Values below 0.1 µg m⁻³ are white.**





While Fig. 2 provides an overview of the mean pattern, daily flow pattern can be quite complicated (e.g. Fig. 3), modulated by monsoonal surges and breaks driven by such factors as convection patterns and tropical cyclone activity. Consequently, many aerosol environments are advected into the study area from around the region, with more variability around Luzon. As outlined in Hilario et al. (2021) and Reid et al. (2023), CAMP²Ex/PISTON was able to monitor MC burning and pollution, Asian pollution, and cleaner marine conditions. For example, off Borneo air was advected well within the southwesterly monsoonal flow, whereas near Luzon, Philippines, air masses originated from all around the region depending on the state/strength of the monsoonal flows. Based on the back-trajectory analysis of Hilario et al. (2021), we segregated P-3 profiles based on their MABL and LFT origin. The dominant aerosol sources for the MABL (~100 m to 400 m) and the free troposphere just above the boundary layer clouds (between 1200-1600 m) are included in Table 1. The average P-3 profiles, excluding cloud sampling, are categorized as: 1) cleaner Western Pacific air advected from the east (WP) on RF 8, 15, and 19; 2) two significant biomass burning outbreaks (BB) that were sampled on consecutive days on either side of the Philippines for RF 6-7, 9-10; 3) other Maritime Continent and Southeast Asia outflow (MC) on RF 1, 3-5; 4) East Asian sources, predominantly from China (EA) on RF 11-14, 16, 17; and 5) mixed and/or indeterminant origins around the Philippines, including the Manila superplume and ship emissions --(Mixed and/or other, MX) on RF 2, 18. Key parameters provided at the two levels for each flight are CO, submicron aerosol volume from the LAS, coarse mode geometric aerosol volume derived from the LAS, and dry light extinction at 532 nm derived from the 3λ nephelometer and PSAP. Average profiles of these and other key constituents are shown in Fig. 4, calculated as the mean of all offshore datapoints by flight by the indicated RF numbers, as the mean of flight means. Included are a) fine and coarse mode volume concentrations as calculated by the LAS and APS; b) non-volatile and volatile condensation nuclei concentration; c) AMS organics and sulfate mass concentrations; and d) dry and liquid water light extinction at 532 nm calculated from the nephelometers, PSAP, and hygroscopicity measurements. Also included is "excess" CO, where the lowest value of CO measured during the mission (70 ppb) is subtracted from the mean profiles. Mission-wide, flight averaged peak particle indicators are always below 1200 m and often in the lowest 400 m (i.e. in the MABL's mixed layer). Although local exceptions exist as shown below in lidar data and such as for nucleation processes associated with free troposphere cloud detrainment (Xiao et al., 2023). Specific regimes are discussed in the subsections below with the regions of flight concentration listed by flight numbers on Fig. 2.



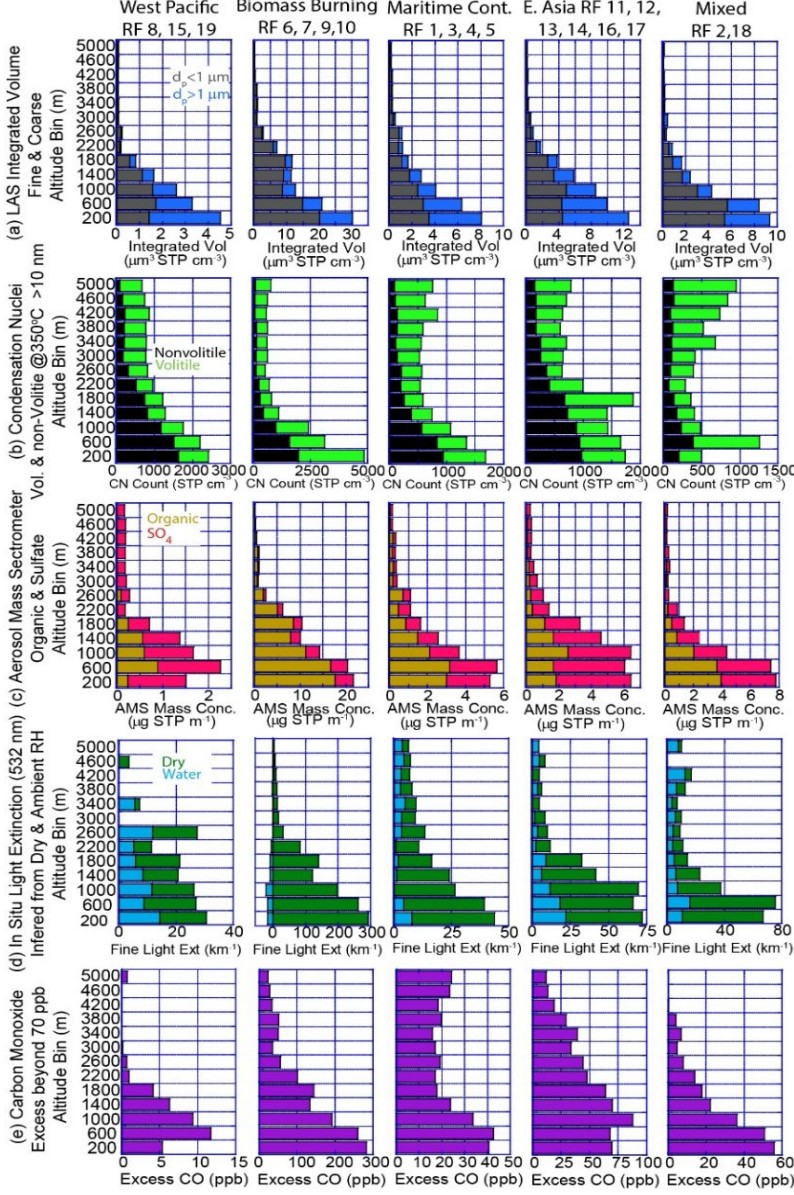

**Figure 4. Mission average profiles of key aerosol parameters based on their primary marine boundary layer source. Included are columns (left to right) more pristine marine sources from the Western Pacific; two major biomass burning outbreaks; Sources from the Maritime Continent and Southeast Asia; East Asia; and complex mixtures of local and transported pollution. Parameters include (top to bottom), row (a) estimated dry geometric volumes for sub and super microns sampled through the P3s inlet; (b) Nonvolatile core and completely volatile CN concentration for particles >10 nm; (c) AMS inferred submicron organics and sulfate; (d) Fine mode light extinction at 532 nm for dry particles and inferred water derived from the PSAP and dry and ambient nephelometer; (e) Excess CO above 70 ppb.**

19999



## 3.1 Western Pacific

Western Pacific air was well sampled once in the SWM (RF 8) and twice in the NEM (RF 15, 19). Back trajectories from Hilario et al. (2021) suggested that for these cases air masses sampled by the P-3 spent more than 7 days over the ocean originating east of the Philippines. If we consider the Western Pacific as the region's "background environment", it is unsurprising particle volumes/masses/extinctions are overwhelmingly below 2 km in altitude, with more significant coarse mode contributions from sea spray as one approaches the MABL (mixed layer inversion was typically ~450-640 m; see Section 4 and 5). CO was quite low, ranging from 74-77 ppb for all three flights. Given the aircraft inlet's aerodynamic cut point of 5.0 um aerodynamic diameter, these coarse particle volumes are certainly underestimated. A slight fine mode aerosol particle enhancement is found just above the typical MABL inversion height of 450-550 m in the fine mode LAS and AMS sulfate and some organics (sea salt species are not measured). This peak coincides with a modest CO enhancement. At the same time, CN concentrations do not peak as fine particles, with particles with nonvolatile cores also making an increasing fraction of counts towards the surface, suggesting secondary production processes dominating particle sources in the middle free troposphere as discussed in detail by Xiao et al. (2023). All of these findings are suggestive of extraordinarily long-lived aerosol species or secondary production originating from distant anthropogenic sources being embedded in the "background environment" (e.g. see discussion below and Clarke et al. 2013). Fine mode light extinction values are also low, and are 40% contributed by hygroscopic water uptake-again expected for high relative humidity found in the MABL (~70-80% RH at the surface, saturation at the MABL Mixed Layer top). Some indications of occasionally observed detrainment layers are also visible in the free troposphere (e.g., Xiao et al., 2023, Supplement Fig S10).

## 3.2 Biomass Burning

In contrast to the *near* pristine Western Pacific air masses, CAMP²Ex sampled two exceptional Borneo smoke outflow events, each on consecutive days on either side of the Philippines (RF 6-7; 9-10). The 2019 burning season was one of the largest on record, with AOTs seasonally nearing and at time exceeding the mammoth 1998 and 2015 El Nino induced events (Eck et al. 2019; Reid et al. 2023 supplemental materials). Such AOTs have not been observed since 2019 at the writing of this paper. Yet, the 2019 season was marked by neutral but warming El Nino/Southern Oscillation (ESNO) conditions followed by a modern record Indian Ocean Dipole (IOP). The season nevertheless followed the emissions and transport that are typical of previous major events, including strong September through October emissions. Examining Fig. 4, we see as previously noted all major indicators of plume transport (Particle size, AMS, CN, light extinction, CO) placing smoke near the surface with indications of lower free troposphere plumes. Unsurprisingly, smoke was dominated by organic species in the AMS with mass and particle volume overwhelmingly below 1600 m for all cases, from off of Borneo to the east of the Philippines. In comparison, MABL mixed layer heights were on the order of 500 m. On one flight, RF9 (September 16, 2019 local day), an isolated entrainment layer was visible at 1800-2200 m, a case shown in more detail in Section 4.



Also notable is that while these flights monitored exceptional levels of fine mode smoke particles, the super-micron particle volumes were non-negligible. We hypothesize these coarse mode particles come from two sources. First, some coarse mode particles are a natural part of the burning emissions, including dust and ash (Reid et al. 2006). Second, major biomass burning outbreaks are typically part of a monsoon enhancement that exhibit stronger near surface wind speeds and may generate sea
salt particles through white-capping.

One key area of continued investigation by the science team is smoke particle hygroscopicity. In cases of heavy smoke, fine mode particles were measured to have light scattering hygroscopic growth factors less than one (i.e., $f(80)<1$). That is, hydrated particles scatter less than their dry counterparts. This is not an uncommon observation, thought to be a result of possibly chain aggregates collapsing, volatilization effects, and refractive index modification while hydrating (Shingler et al. 2016); Lorenza
et al., 2024). However, as shown in the lidar profiles, overall particles are clearly hygroscopic. There has been much internal discussion as to what such hygroscopicity measurements mean and how they compare to bulk lidar observations (which include total fine and coarse, including any monsoon generated sea salt). The potential for measurement artifacts is also frequently discussed. While reconciliation of these observations is outside the scope of this paper, this topic is discussed further in Section 4 and 5.

**3.3 Other Maritime Continent Observations**

While major biomass burning events were observed toward the end of the SWM, the P-3 flew multiple cases off Borneo when fire prevalence was low (RF 1,3,4,5). These cases are likely a mixture of some burning, and also regional anthropogenic emissions (domestic fuel, industry, petrochemical, ship emissions, etc.; e.g. Hilario et al. 2020). For these cases from the MC, we see a similar pattern of low-level transport as with biomass burning with much lower and more distributed particle
concentrations to ~3000 m. The sulfate fraction is significantly enhanced over biomass burning with slight enhancement in hygroscopicity, demonstrating a mixture of biomass burning and industrial contributions. Further, the coarse mode volume is a larger fraction of particles overall, although the absolute magnitude of coarse mode for the MC cases is half of what was present in the biomass burning cases. Also notable is that for the MC and biomass burning overall, there are elevated CO values for altitudes above where particle concentrations fall off. This can be reasonably explained as a result of long-range
transport from the rest of SE and South Asia (e.g., Reid et al. 2015) and detrainment of aerosol scavenged air.

**3.4 East Asia**

In comparison to biomass burning and other Maritime Continent emissions, the East Asian sourced flights showed the first indications of a significant land plume transport phenomenology. East Asian sources were sampled on RF 11, 12, 13, 14, 16, 17 around Luzon and they showed higher particle concentrations above the mixed layer indicative of land plume formation.
These flights are the only ones from land sources where sulfate dominates fine mode aerosol particle masses over organics.



As one would then expect, higher hygroscopicity was exhibited from other land sources as well. Also notable are CO and volatile CN enhancements in the lower free troposphere.

## 3.5 Mixed Environments

Finally, two flights can be categorized as mixed environments (RF2 and 18), that were an amalgam of local Philippine sources,
including the Manila Super Plume and well-aged Asian and Maritime Continent sources including ship emissions. These mixed cases were just that, variable in chemistry and vertical profile, although nevertheless dominated by particles <3 km in altitude but with high CN volatility fraction and an enhancement in CN concentrations. Notable for both cases are the enhancements in CN concentrations in or just above the MBL as well as higher CN concentrations aloft. For RF2, this is in part due to observed ship emissions in the area. For RF18, urban emissions in the presence of enhanced radiation led to explosive
nucleation (e.g. Reid et al. 2016).

## 4 Results 2: Example regional aerosol environment as monitored from the P-3 HSRL-2

Of the 19 CAMP²Ex research flights, several stand out as having easily describable maritime environments that demonstrate commonly observed aerosol features. Here we examine a) the background marine cases of the Western Pacific (RF15 and 19); b) the biomass burning smoke outbreaks (RF 6, 9, and 10); and c) East Asian pollution (RF 17). Excluded in these cases is an
evaluation of the many multi-spectral lidar products, as well as more complex environments, mixed maritime continent outflow, and local pollution of Philippines; these are all topics that are quite nuanced and require separate, more in-depth investigations. For this section, we extracted cases with good P-3 HSRL-2 viewing conditions to the surface from altitude and successful 532 nm light extinction in the vicinity of dropsondes and in situ profile data that typified the nature of aerosol layering phenomenon (Table 2). Even so, because of the high frequency variability shown here and in Section 5, it is quite
difficult to perform true closure between the lidar profiles, sondes and in situ profiles. Nevertheless, these cases represent more idealized conditions of different classes of aerosol environments. Provided in Fig. 5 are 30-second average aerosol backscatter clean marine and polluted cases, in (a) and (b) respectively (the average of three consecutive 10 second profiles from the standard LaRC HSRL-2 product). Corresponding light extinction and lidar ratio products for the same 30 second product window are shown in (c)/(d) and (e)/(f), respectively. It should be remembered that lidar extinction and lidar ratio in the HSRL-
2 archived data products are computed on a 60-s running average reported on a 10-s grid, making a 30-second product average shown here to have an effective smoothing of 90 s (~114 km horizontal center-weighted average assuming the P-3 flying at 300 kts). As the clean marine conditions are near the noise threshold for extinction and lidar ratio, lidar products in Fig. 5 have been further boxcar smoothed to 90 m in the vertical, making a 360 m center-weighted product. Finally, the relative humidity fields from nearby dropsondes are provided in (g) and (h).


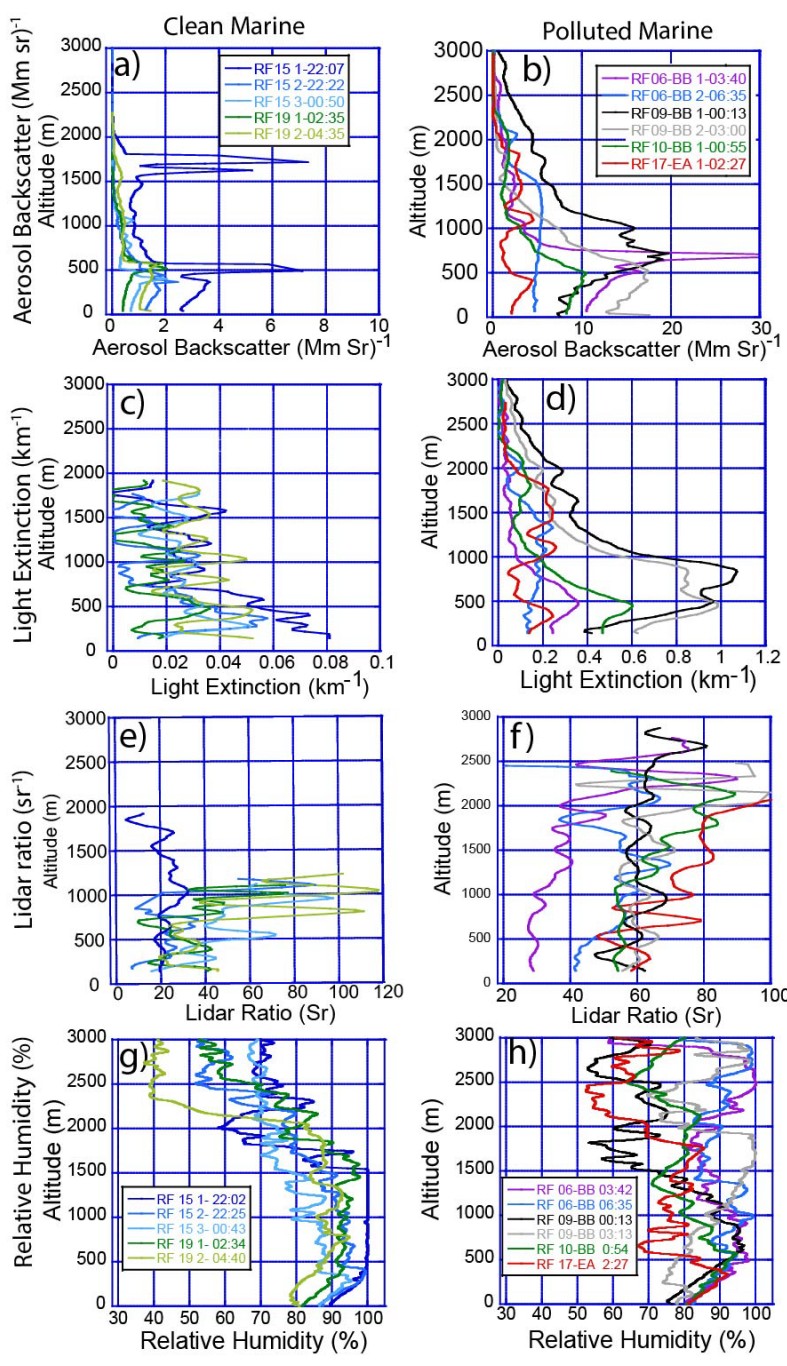

**Figure 5. Example LaRC HSRL-2 profiles at 532 nm of common phenomena for clean marine (a, c, e) and polluted (b, d, f) marine conditions. Included are 30 second (~4.5 km) aerosol backscatter, and extracted 1.5 minute (~14 km) aerosol extinction and lidar ratio with additional 90 m boxcar smoothing. Also included are nearby dropsonde releases**

**((g) and (h)).**



### 4.1 NWTP Background Marine

We begin the analysis of P-3 observations of "clean background marine" environments that often come to mind as being representative of the Western Pacific. As this is the first example shown in Section 4, we go into more detail here than subsequent environments. As noted in Clarke et al. (1996; 2003; 2013), Reid et al. (2022), and Section 3.1 above, the idea of "background" or "pristine" marine can be problematic as there is often some terrestrial influence on aerosol populations in the remotest of oceans. Indeed, enhanced CO, as an indicator of anthropogenic influence, was found just above the MABL (Fig. 4). Nevertheless, after reviewing flight data, two flights immediately stand out as being closest to nominal "background marine" conditions observed during CAMP²Ex: RF 15 (28 Sept, 2019 local date) and 19 (5 Oct, 2019 local date). RF8 is excluded as it has some orographic influence of small islands along the aircraft track. RF 15 and 19 were conducted post monsoon transition and these flights experienced excellent fair-weather subtropical conditions while capturing key MABL and cloud detrainment phenomena. Visible images with flight tracks and 925 hPa NAVGEM winds are provided in Fig. 6(a; Terra MODIS) and (b; SNPP VIIRS) for RF15 and RF19 cases, respectively. Subsequent panels provide P-3 forward camera images and HSRL-2 aerosol backscatter curtains. It is clear from the Fig. 5 images that for "fair weather marine" conditions there are numerous macroscopic MABL cloud organization or "textures" for similarly low to moderate wind conditions. Back trajectories from Hilario et al. (2021) for these flights reveal airmasses originating from the Western Pacific. (Table 2) has estimated AOT at 532 nm ($t_{532}$) from the HSRL-2 processing algorithm ranging from ~0.03 to 0.09. These are within one standard deviation of what sun photometers observe in typical remote ocean (e.g. Reid et al. 2022). MABL inversion height estimates varied in the 450 to 675 m range.

RF 15 (Sept. 28, 2019 local day) represented a period of easterly MABL winds ahead of Typhoon Mitag, which was propagating into the northeast quadrant of our operations area (Fig. 6(a)). The P-3 sampled an array of macroscopic cumulus cloud organizational structures of cloud properties before sunrise to mid-morning. The flight started with a pre-dawn Aeolus satellite wind lidar underpass after which the P-3 sampled two distinct regimes: 1) Between points 1 and 2, across an area of some crosswind oriented weak cloud development; and 2) A similar length line west of Point 3 in an area of scattered MABL clouds, largely subpixel in the MODIS imagery. A third area of investigation by the P-3 to the north was similar to the second and thus not dealt with here. In the two regimes shown in Fig. 6(a), MABL winds were largely northeasterly to easterly, with cloud features and macroscopic textures aligned north-south/cross-wind. NAVGEM and dropsondes suggest convergence across the convective line, with winds ~7.5 m/s @25° on the eastern side below 850 hPa, 5 m/s @50° on the western side below 850 hPa, and diminishing winds at and beyond 700 hPa. Examining geostationary data, this cloud band was shed from TC Mitag 24 hours earlier while it was in a tropical depression stage. Moisture profiles (Fig. 5(g)) show the typical increase in RH in the ML from the surface to the inversion base at ~400 m, followed by a moist layer to 1700 m, and then some drying aloft (as is often observed in tropical maritime regions). For the segment between points 1 and 2, the P-3 cut across a feature





of *cumulus mediocris* (Cu med) with cloud top heights to 2-3 km and thin detraining *altocumulus* (Ac) clouds, associated aerosol layers, and outflow remnants on the western side between 500 m and 1500-1800 m (Fig. 6 (a) marker 1; Fig. 6 (c) &

(f)). As noted in Table 2, the western "downwind" side of this convective area had HSRL-2 derived 532 nm AOTs ($t_{532}$) of 0.09 versus 0.06 on the eastern side. The western side AOT and backscatter enhancement is consistent with detrainment, although long range continental transport from Asia cannot be completely ruled out. Nevertheless, small areas of enhanced particulate backscatter (i.e., so called "halos"; Radke and Hobbs, 1991) are occasionally visible around clouds on the eastern side of the convective line.

The second RF15 domain around Point 3 had appreciably less convective development than RF15a. The region hosted small fields of *Cumulus Humilis* (Cu hum) with limited development, largely 100-200 m deep, with one cloud top height observed reaching 1 km (Fig. 6 (d) and (g)). And yet, dropsonde RH fields show only a slightly drier environment just above the MABL than in the convergence zone between points 1 and 2. This airmass originated from Mitag just behind the convective line sampled between Points 1 and 2. Consequently, the derived $\tau_{532}$ product generated as part of the standard LaRC processing

was low and matched Point 2 with a value of 0.06. Even though convective development was less than the other domain, about 1/3 of light extinction and aerosol backscatter was below cloud base at approximately 425 m, with limited detrainment to 1 km. What is highly notable, however, is that one plume was observed between 800 and 1400 m at the ~100 km marker. This is likely a cloud halo/residual from an evaporating cloud.

The second example day of a Western Pacific marine environment, RF19, differentiates itself from RF15 in the sampled
region's cloud organization and development associated with flow around a mesoscale convective system (MCS; Fig. 6(b) and (e)). Like RF15, near surface winds were roughly in the 6-8 m/s range, with areas of *Cu Hu* and *Cu Med* to 3 km in height with prevalent Ac (Fig. 6 (e), (h) and (i)). However, for these cases the macroscopic cloud pattern is along wind instead of crosswind, an indicator of MABL roll structures (e.g. *Moeng and Sullivan 1994; Salesky et al. 2017; Park et al.* 2022). Moisture profiles were slightly drier in the ML than the RF15 case, especially for the second profile. RF19 sampled in both the cross and along
wind directions (versus only cross wind in for RF15). $t_{532}$ was also similar to RF15 (~0.06-0.09; Table 2) with lidar ratios at ~30 sr in the mixed layer, and increasing with height (Fig. 5).




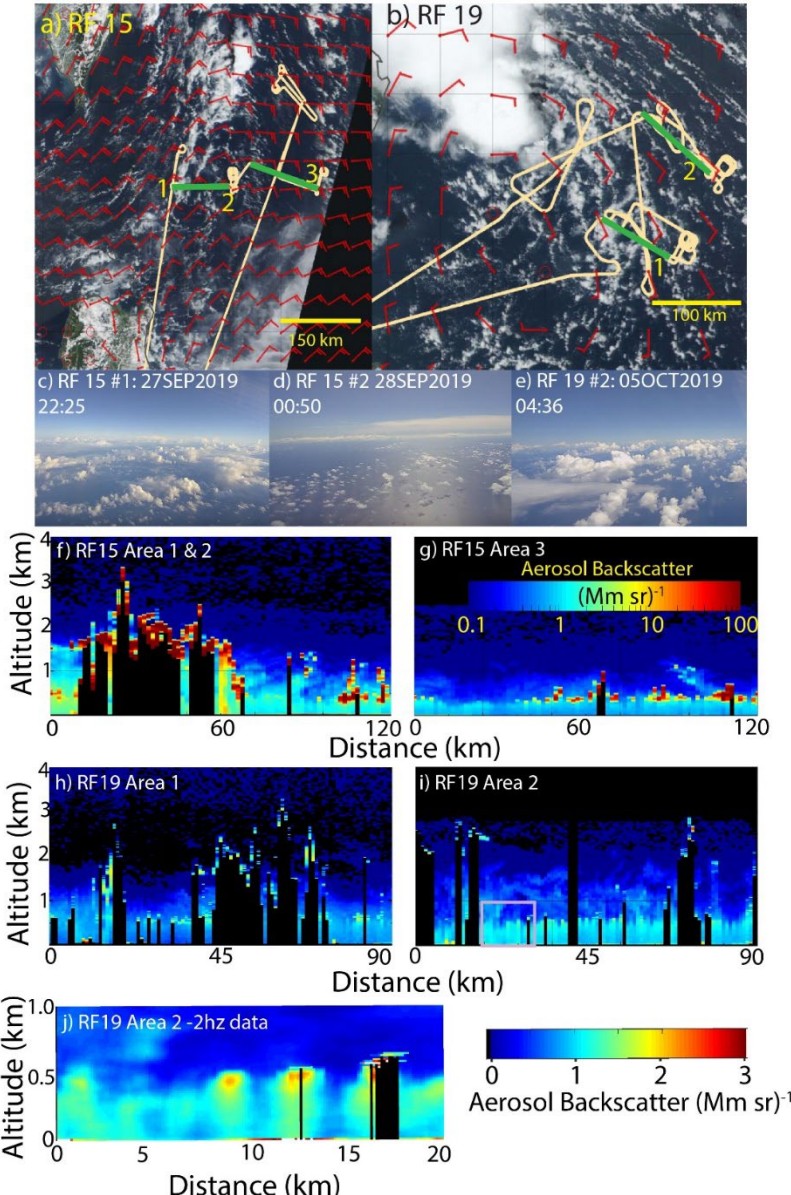

**Figure 6. HSRL-2 aerosol backscatter profiles for two research flights focusing on background marine conditions (RF 15, takeoff 27 Sep, 2019; RF19 takeoff Oct 5, 2019). a) & b), Terra MODIS RGB with P3 flight tracks and curtain segments labeled; (c)-(e) selected forward camera at the start of lidar segments used here;(f)-(i) HSRL-2 calibrated aerosol backscatter with cloud screening for flight segments marked on (a) and (b)-all with colorbar noted in (g); and (j) 2 Hz zoom of the box in in (i) with its own linear color bar. Blacked out areas of lidar profiles are due to the presence of opaque clouds.**



These cases provide us with some spatial perspectives of clean marine conditions. Both flights sampled areas low in AOT. While the HSRL-2 measurements of AOT have compared very well with Sun photometer AOT measurements, the instrument is near the signal-to-noise extinction derivation capabilities for the airborne HSRL-2 (Sawamura et al. 2017). Indeed, examining 30-second average aerosol backscatter profiles (Fig. 5(a)), we easily observe well-known characteristics of a boundary layer. Backscatter increases in the mixed layer due to adiabatic cooling and likewise increases in RH, as demonstrated

in the associated dropsondes. When backscatter reaches a maximum coincident with potential temperature at the top of the mixed layer, roughly at cloud base, backscatter then also asymptotically falls with altitude with punctuations of thin aerosol layers and Ac clouds. However, light extinction and lidar ratio (Fig. 5 (c) and (e)) are difficult to interpret, even with the additional averaging imposed.

Despite the noise in the derived extinction, we can still estimate the nature of ambient light extinction through the aerosol

backscatter product. Within the MABL, backscatter from these background days can vary by a factor of two. With the exception of RF15 Point 1 (e.g. ahead of the weak convective line), backscatter values are all fairly similar above the MABL. While aerosol backscatter is dominated by the MABL, as is aerosol mass as shown Fig. 4, the fainter aerosol backscatter in the free troposphere integrates spatially to appreciable values. Indeed, overall, 25-45% of the estimated extinction based on an assumed constant lidar ratio for these cases is within the MABL Mixed Layer (Table 2). If we assume the increasing lidar ratio

with altitude is also due to the increase in RH near the top of the MBL (e.g. Fig. 5(e)), then this fraction diminishes further-although establishing the veracity of the derived lidar ratio under these clean conditions requires a study in and of its own. Nevertheless, the role of spatially extensive but low extinction aerosol environments has been problematic to the lidar community for some time. Noteworthy is the exclusion of such environments in CALIOP retrievals with a subsequent retrieval bias (e.g. Toth et al. 2018).

Figure 6 also shows that overall the height of pronounced aerosol backscatter is often, but not always, related to the height of local cloud tops. This is clearly visible in both RF15 cases, as well as RF19 area 2. Indeed, even with cloud tops to 3 km in RF15b, a clear delineation in aerosol backscatter is visible. And yet, for RF19 area 1 the opposite is true, as there is a minimum in aerosol backscatter between 2 and 3 km. Such collocated enhancements or dearth of aerosol particles may well be related to changing natures of cloud to detrainment for precipitating versus non-precipitating cells-either through scavenging or

through differences in the detrainment processes themselves.

Fine detrainment layer structure and cloud halos are also clearly visible in the individual profiles (Fig. 5) and curtain plots (Fig. 6(f)-(i)). Thin aerosol layers may be produced by nearby cumulus cloud development, or generated by deep convection and differential advection further afield. The nature of detrained aerosol layers in and around cumulus clouds is quite apparent with the HSRL-2 curtains in Fig. 6(f)-(i). The deeper Cu fields for RF15 Area 1 have more visible detrainment layers aloft,



while cloud fields with limited vertical development (e.g. RF15 Area 3) do not.   Nevertheless one isolated cloud residual is visible at 1 km in altitude. However, this is not a universal observation, as deeper convection was present in RF19a to the same heights as RF15 Area 1, but did not show as strong detrainment features. One hypothesis explaining this difference is related to the forcing of the cloud fields. The cross-flow cloud feature in RF15a may be forced by slight near-surface convergence, leading to net detrainment aloft. Whereas for the RF19a case, cloud streets are a result of shear and differences in buoyancy

(Lemone, 1973; Moeng and Sullivan, 1994). And yet, for RF19 Area 2, Ac were prevalent in that RF19 area, as were more visible detrainment layers. One notable difference between RF19a&b is sampling. The RF 19 Area 1 sampled more cross wind, whereas RF19 area 2 was more along wind. As this has implications for MABL sampling (e.g. Park et. al, 2022), it is probably fair to hypothesize along wind/cross wind sampling bias plays a role in MABL cloud detrainment studies as well. Other factors potentially impacting detrainment, and hence the formation of Ac and associated aerosol layers, include environmental static

stability and relative humidity, the vertical structure of cloud buoyancy, in-cloud vertical pressure gradients, and evaporation along cloud edges (e.g., Moser and Lasher-Trapp 2017; Savre 2022; Morrison 2025), and should be further explored.

Finally, in addition to MABL cloud detrainment variability, there is also significant vertical structure in the mixing layer itself, shown in figures such as Fig. 5. As noted above, we expect an increase in aerosol backscatter with height in the mixed layer due to simple adiabatic cooling, leading to increases in RH, and hence hygroscopic growth. However, updrafts under clouds

with subsequent in-between drier downdrafts (e.g. Reid et al. 2017) and associated MABL roll structures (Park et al, 2022) also result in alternating higher and lower scattering columns. Examining the 10-second data in Fig. 6 (g) and (i) in particular, we can see horizontal oscillation in backscatter in the lowest 500 m. Given the P-3 was flying at 170 m/s, a 10-second sample is 1.7 km, not too far off of the common 2.5-3 km distance between the roll structures comprising  cloud streets. If we plot backscatter at the instrument native resolution of 0.5 seconds (noting that 10 seconds is still considered the full independent

data rate of the instrument, Fig. 6(j)), the enhanced backscatter column under clouds becomes much clearer. Oscillating areas of enhanced backscatter of about a factor of 2 extend around the cloud to the surface. Although the independent signal sampling period is 10 seconds, the symmetry of these features around the clouds suggests they are physical and not an electronic signal problem. We surmise that these features are related to updrafts and the nonlinear nature of a hygroscopic growth curve and vertical exchange of air under and between clouds (e.g. as in the terrestrial PBL described in Kunkel et al., 1977 and Reid et

al. 2017). Such variability is examined in much more detail in Section 5.

## 4.2 Biomass burning outbreaks

In this subsection, we move from the very cleanest environments sampled by the P-3 during CAMP[2]Ex to the most polluted. The P-3 sampled the 2019 season's two major SWM smoke outbreaks, both related to monsoonal enhancements associated with tropical cyclone development in the NWTP-a common source of monsoonal perturbation (Reid et al. 2012). The first

enhancement was driven by the passage of Typhoon Lingling from the east of the Philippines up through north of Taiwan and into North Korea. Smoke from Central Kalimantan, Borneo was sampled just to the west of Luzon on RF6 after ~1600 km of



transport (Fig. 7(a) local day 7 Sept 2023) within a well-defined wind enhancement of 12 m/s. $t_{532}$ from the HSRL-2 was on the order of ~0.3 (Table 2). This plume was transported in active warm convection with cloud tops at 3-5 km transitioning to deep convection due to orographic enhancement along the Luzon coast. A second smoke outbreak occurred 9 days later by the

influence of multiple tropical disturbances that eventually formed into Typhoon Tapah. This outbreak was sampled on subsequent days: RF9 sampling 150 km north of Borneo where $t_{532}$ were measured >1 after ~1000 km of transport from Central Kalimantan (Fig. 7(b) local day 16 Sept 2023); and RF 10 (Fig. 7(c) local day 17 Sept 2023) sampling just east of Luzon with $t_{532}$ measured at 0.6 after an additional 1000 km of transport. While the RF9 and 10 were roughly one day apart in transport given the MBL winds at 10 m/s, we are reluctant to call these two flights a true Lagrangian pair. Nevertheless, some

consistencies and differences are indicative of evolution processes.

In contrast to the background marine conditions, biomass burning outbreaks sampled during CAMP$^2$Ex were well above signal to noise limitations for the HSRL-2 for extinction measurements. Figure 5(b) & (d) show extinction profiles that largely match aerosol backscatter structure quite well, although smoothed due to its 300 m native sampling window. Lidar ratios are also more stable than marine counterparts (Fig. 4(f)) and vertical variability can be resolved-albeit nevertheless with some potential

noise. In all but one case (RF6 #2, 6:35; see below), we find the typical aerosol backscatter profile peaking at the top of the 450-650 m high mixed layer. Also embedded in the mean profiles are many geometrically thin, but high backscatter aerosol layers that can persist for 10s to 100s of kilometers as well as what appears to be other detrainment events and residual cloud halos. Frequent oscillations at the top of the smoke layers observed in RF9 are indicative of gravity waves (Fig 7i).

Examining these three flights together, there are notable similarities and a few key differences between lidar backscatter

profiles shown for the RF 6 and RF 9-10 cases. In all cases, over 98% of measured aerosol backscatter was at altitudes below 3 km (Fig. 5) and subjected to low-level transport. These cases also demonstrate the nature of cloud processing and detrainment of aerosol particles in the presence of vertical wind shear. The RF 6 profiles have the same $t_{532}$ at ~0.3, but exhibit vastly different profile distributions. RF6 #1 being MABL dominated, and RF6 #2 showing one of the few cases having a very low fraction of extinction in the mixed layer (28%). These profiles for RF6&7 were taken within and just outside the area of

convection associated with the monsoon enhancement (Fig. 7(a)). Track RF6 #1 with its MBL dominated smoke transport (note red arrow) was collected in a cloud free slot in between convective lines with prevalent *Ac* and noticeable detrainment to 3 km (Fig. 7 (d) & (g)). Even near the surface, there is considerable variability in profiles and layer strengths. In contrast, track RF6 b was on the south side of the monsoon enhancement cutting northwards into convection. In Fig. 7 (h), we can clearly see a shift from an aerosol environment dominated by smoke above the MABL on the southeast side of the main

convective area (note red arrow, distance marker 0-10 km) to profiles within the convection (distance of 65 km) that are more indicative of MABL transport as was seen in RF6 a. We initially surmised that the smoke layer at 1200 m may be a result of detrainment and shear from the eastern side of monsoonal enhancement transporting the smoke. However, this shear hypothesis is not supported by the available dropsonde data. Alternatively, this enhancement at RF 6 #2 may not be an enhancement at





all, rather, smoke may be efficiently scavenged in the convective area, and thus the lower free troposphere is left with many
optically thin detrainment layers. Or, as discussed in Section 4.4 may be a result of large scale cold pool lifting.

The pair of flights on RF9 and 10 are notable in that unlike RF6, they were not embedded in a region of highly active convection, but rather in more scattered congestus and deep convection. Long transects at altitude on RF9 allowed for good HSRL-2 viewing conditions to fully sample the environment. Significant variability in aerosol profiles is visible in the lowest kilometer, with multiple aerosol layers visible that are 30-100 km in length but only a few hundred meters in depth. While the
smoke layer is largely capped at 3 km, very thin aerosol layers were visible up to 4.5 km, the approximate 0°C level. After an additional 1000 km of transport, the center of the plume was observed again the next day during RF10. While some detrainment is visible to 3 km, the majority of extinction is within the mixed layer. This is perhaps the mission's best example of how a change in vertical profile can be a result of slight changes in wind shear. Winds from the surface were 13 m/s from the southwest, while winds aloft increased to 15 m/s west-southeast with smoke being preferentially advected more eastward.

Like the marine environment, biomass burning smoke cases all exhibited systematically increasing lidar ratios with altitude (Fig. 5 (f)). This results in noticeable shifts in light extinction weighting upwards relative to aerosol backscatter. From a lidar data interpretation point of view, these changes can have significant implications for estimating aerosol budget and transport. In the case of RF9 #1 00:13, this increase in lidar ratio shifts the maximum extinction from the top of the mixed layer to just above. Thin aerosol layer enhancements above MBL are also enhanced by the growth of lidar ratio with height. A second area
of interest is notable differences in the lidar ratios for the two monsoon enhancement events. Typically, smoke has higher lidar ratios ~60 sr or above (Burton et al. 2012), as was seen for the RF 9 and 10 flights of local day 16-17 September 2019. However, for the first monsoon enhancement of Sept 7 (RF6) case, lidar ratios are a third to one-half of that. Surface winds for RF6 were ~12 m/s and over a long maritime fetch; perhaps lower lidar ratio of 30-40 sr are a result of mixing with coarse mode sea salt. Yet this explanation is not entirely satisfactory, as the RF 9 and 10 pair exhibited high winds too and the ratio
of fine to coarse mode volume in the MABL from the LAS is also the same at 0.36 (Table 1). A second possibility is the lower lidar ratios for RF6 are simply a result of the lower overall AOTs for that event, and hence a greater coarse/giant mode fraction that is not sampled in the P-3's inlets. Lastly, we pose that perhaps these lower lidar ratios in part have to do with differences in the nature of cloud processing of smoke for the airmass sampled in RF6. Indeed, this region was quite active with warm multi-thermal convection (see analysis in Reid et al. 2023). Nevertheless, this difference is highlighted here to demonstrate
that even for what one would classify as a "major biomass burning event" interpretation of the lidar data is not straightforward. Higher order analysis of this comparison is underway.



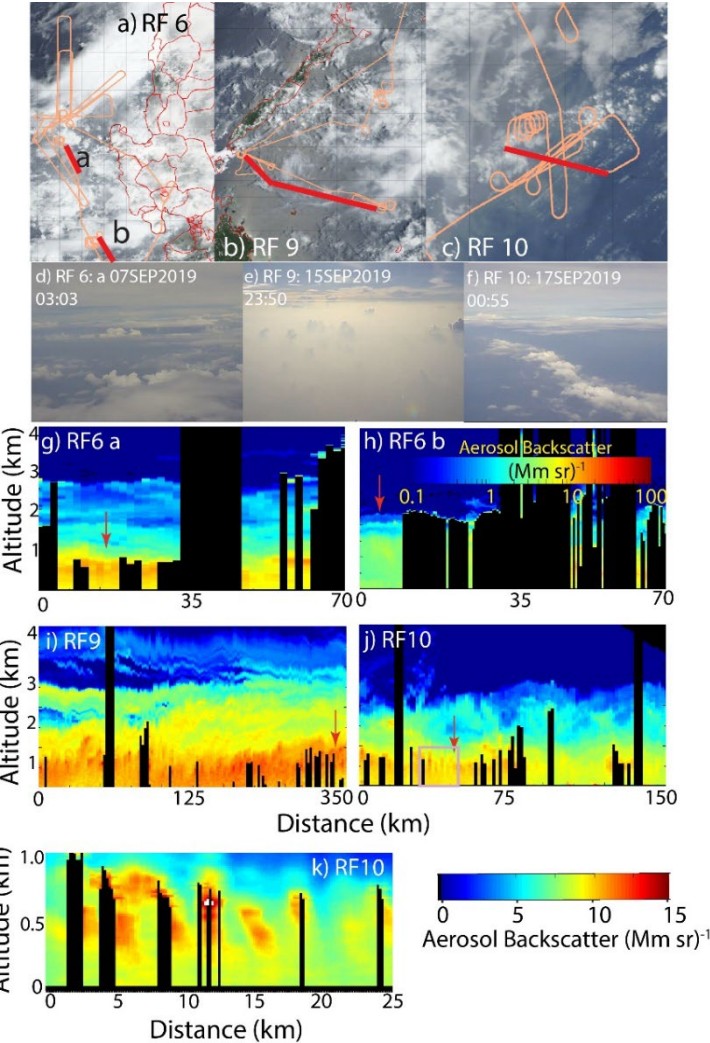

**Figure 7. Same as Figure 4 but for HSRL-2 aerosol backscatter profiles for three research flights focusing on major biomass burning smoke outbreaks (RF 6, takeoff 06 Sep, 2019; RF9 takeoff 15 Sep 2019, 2019; and RF 10 takeoff 16 Sep, 2019 ). a), b), & c) Terra MODIS RGB with P3 flight tracks and curtain segments labeled; d)-f) selected forward camera at the start of lidar segments used here. g)-j) HSRL-2 calibrated aerosol backscatter with cloud screening for flight segments marked on a)-c). Marked in red arrows on (g) and (h) is a contrast in notable differences in the aerosol vertical distribution inside and outside the convective region. Likewise arrows in (i) and (j) notable differences between boundary layer profiles sampled just off of Borneo and after 1000 km of transport to the north east demonstrating how shear results in separating the plume into a MABL and lower free troposphere plume. Finally, in (k) is a zoomed 0.5 second sample of the purple box in (j) showing individual role feature induced backscatter enhancements slanted due to the presence of wind shear.**





Increasing values in aerosol backscatter and extinction profiles indicate increases in extinction with height in the MABL
suggesting increasing water-uptake due to enhanced particle hygroscopicity. This is contrary to onboard in situ hygroscopicity measurements in the LARGE package visible in Fig. 4 and noted in Reid et al. (2023) as needing further investigation. Possible reasons for this difference may be physical, such as the role of sea salt (i.e. hygroscopicity measurements of only those penetrating particles through the P-3 inlet versus the clearly hygroscopic nature of particles within the ambient lidar beam) or some measurement artifact related to particle drying and dehydration process in the instrumentation. This is discussed to some
extent by Lorenzo et al. (2025) and commented on further in Section 5

Lastly, notable are individual updrafts and cloud halos as in the clean marine case. With surface winds in monsoon enhancements at 12 m/s, convective roll structures are also visible in 2-Hz (90 m) aerosol backscatter data (Fig. 7(k)). As with background marine, oscillating pillars of higher and lower aerosol backscatter are observed ~2-3 km apart. However, these oscillations are lower in magnitude than the maritime counterparts-perhaps due to less active convection or lower
hygroscopicity of the smoke particles. Given the higher winds associated with monsoon enhancements, there is notable evidence of wind shear-aerosol backscatter in tilted convective lines rather than vertical pillars.

### 4.3 Asian Air Pollution

In comparison to heavy biomass burning smoke, RF17 sampled northwest of Luzon the most significant Asian pollution case advected into the region on 2 Oct 2019 (Fig. 8(a)). Post-monsoon, winds over the northern South China and Philippine Sea
were quite light and variable (Table 2). Leading up to this flight, pollution from China, including from the Pearl River Delta region was transported to the southeast towards Luzon. RF17 occurred right between a shift in the winds to more northerly conditions, with speeds being variable and <1 m/s winds from the surface to 2 km. North of Luzon, there were fields of very shallow Cu Hum with thicknesses of 50-200 m (Fig. 8(b)), although slightly deeper convection was in the vicinity to the south. RF17 fits a land plume model of stratified aerosol features (Fig. 8(c)) with high sulfate to organic ratios (Fig. 4). Above the
typical mixed layer peak in aerosol backscatter at 450 m, a broader land plume is visible from 800-2000 m. Embedded are aerosol backscatter enhancements of just a few hundred meters deep, but up to 50 km long. Lidar ratios for this pollution case were even higher than for biomass burning, being over 60 sr at the surface and increasing to over 80 sr by mid plume (e.g. Fig. 5(f)). Examining the 2-Hz data (Fig. 8(d)), we do not find strong updrafts as in the earlier cases of biomass burning or marine conditions-understandable given the very low wind speeds and most limited cloud development. Nevertheless, individual cloud
halos are still visible with less scattering downdrafts between clouds.





**Figure 8. Sames as Figure 4 but for HSRL-2 aerosol backscatter profiles for the RF 17 research flight focusing on air pollution from East Asia. (Take-off 01 Oct, 2019). a) Terra MODIS RGB with P3 flight tracks and curtain segments labeled; b) forward camera at the start of lidar segments used here. c) HSRL-2 calibrated aerosol backscatter with cloud screening for the flight segments marked on a).**





### 4.4 Cold Pool

The final airborne case presented here is the sampling of an evolving cold pool on RF7 around the September 9, 2019 1:00-4:00 Z timeframe. Unlike previous cases where the P-3 provided large scale surveys of regional environments, this cold pool case demonstrates perhaps CAMP²Ex's best example of P-3 sampling an evolving environment. Describing the complexity of this case with the P-3's remote sensors and dropsondes is far outside the scope of this paper and is covered in much more detail in a future work. However, in the context of this CAMP²Ex's aerosol "animals in the zoo" paper it is worth giving a brief overview here.

After RF7 takeoff, the P-3 was directed to the northeast of Luzon where an isolated thunderstorm began forming at 0:00Z. After arriving at 01:00Z, the P-3 surveyed cloud features around thunderstorm outflow, followed by a 50 km east-west cross section along a 10 km wide clear air slot above the cold pool that began to develop between the storm core and cold pool's leading edge due to the outward propagation of the cold pool itself (Fig. 9 (a), (c)). Forty minutes later the cold pool propagated another 10 km and the P-3 performed a northwest to southeast cross section. Given the complexity of the scene, here we use the 0.5-second HSRL-2 aerosol backscatter for the two transects (Fig. 9 (e) and (f)) as well as estimated surface wind retrievals from the Advanced Microwave Precipitation Radiometer (AMPR; Amiot et al., 2021; Lang et al., 2021) microwave data for context (Fig. 9 (g) and (h)). Dropsondes were scattered over the region.

Overall, the RF 7 observations conceptually match an almost idealized cold pool case (Tompkins 2001; Drager and van den Heever 2017). Geostationary imagery estimated propagation speed of the arc cloud away from the storm center was ~5-6 m/s. For the P-3's first crossing at 1:49Z, the arc cloud tops reached ~ 2 km (distance 8 km on transect), with a congestus to 3.5 km in the middle of the cold pool (distance 35 km) that from satellite imagery may have been from a secondary pulse from the storm. Retrieved surface winds from AMPR qualitatively show significant increases and heterogeneity under the early stages of the arc cloud, rapidly diminishing after it weakened significantly (Fig. 9(g) and (h), respectively). Optically thin altocumulus clouds are also visible in the HSRL-2 as well as the forward and downward camera images. Forty minutes later for the second transect, the cold pool arc cloud had visibly started to dissipate, with cloud tops at only 1 km. Scattered alto cumulus remain visible in the HSRL-2 and video footage.

What makes this case so useful in the context of this paper is that these observations provide more of a mechanistic view of the elevated aerosol layers observed leaving the convective outflow in the RF6 b case provided in Fig. 7. Nominally, our working hypothesis is that smoke is transported long distances in and around the MABL. With the help of vertical wind shear, convectively pumped smoke can be transported in different horizontal directions near the surface. Here, we can observe not just cumulus cloud convective pumping, but the injection of cleaner free troposphere air into the MABL by thunderstorm downdrafts. Indeed, outside the cold pools we find 532 nm AOTs at 0.14-0.18 reducing to 0.05 within (Table 2). We see high aerosol backscatter outside the cold pools (distance <5 km for 1:39 case, and >30 km for 02:19 case) with AOTs on the order



of 0.14-0.18 (Table 2). Behind the cold pool, however, cool density current flow pushed the aerosol layer upwards creating an elevated layer and overall diminished AOTs to ~0.05. Additional scavenging within the arc cloud may also further reduce aerosol backscatter. Noteworthy is how detraining altocumulus clouds are associated with the tops of aerosol layers, as described in terrestrial counterparts by Reid et al. (2019). This lifting of aerosol particles may also explain the enhanced smoke layers aloft for the RF6 biomass burning case. Also notable is enhanced near surface aerosol backscatter immediately behind the arc cloud in Fig. 9(e), we hypothesize due to wind generated sea spray associated with the arc cloud. This is discussed further for a *Sally Ride* case in Section 5.2.4. Finally, complex interactions between aerosol loading, precipitation amounts and size distributions, environmental conditions, cold pool intensity, and ultimately cold pool lofting of particles, have been noted in a number of idealized modeling studies (e.g., Grant and van den Heever, 2015). These co-located CAMP²Ex observations of aerosols and cold pools will assist in further unravelling these interactions.







**Figure 9. Similar as Figure 4 but for the cold pool case on RF 7 around 09 September, 2019 for two flight segments for 01:39-1:24 Z and 2:19-2:24 Z. (a), (b) Satellite image with path corresponding to 0 to 50 km; c), d) forward camera; (e), (f) 2 hz HSRL-2 aerosol backscatter; (g), (h)20 hz microwave derived wind speed. Marked under (g) and are where the P3 is outside and inside the cold pool, as well as the location of a potential secondary cold pool forming.**



## 5   Results 3: PBL and Vertical Variability from the *Sally Ride* HSRL

The R/V *Sally Ride* with its upward pointing UW-HSRL gives a different perspective from the P-3's downward pointing
HSRL2. While the NASA P-3 provides an excellent platform for surveying a region at any given time and is configured to provide estimates of the lidar ratio in the MABL, the near-continuous *Sally Ride* measurements observe aerosol vertical distribution within the MABL and MABL variability. The HSRL provides calibrated backscatter and depolarization to within 150 meters of the instrument, however retrievals of extinction and lidar ratio are problematic for the up-looking observations due to the receiver being out of focus close to the instrument (<5km). Further, the *Sally Ride*'s strategic position has the
advantage of long-term monitoring just upwind of maximum convection around the NWTP monsoonal trough, with the disadvantage of westerly shear advecting upstream smoke above the MABL to the east (e.g., Section 3). This said, smoke aloft is then much more likely to have been placed there by more local convection. In this section, we examine the time series data and cloud related perturbations of aerosol backscatter.

### 5.1 Period Overview

*Sally Ride* started collecting lidar data on September 4, 2019 on its way from Taiwan and was on station September 7th in the NWTP just at the start of the first of two major Indonesian smoke outbreaks. Figure 10 provides an overview of cloud and precipitation screened aerosol backscatter taken over the course of the *Sally Ride* deployment along with bow mast meteorology and radiosonde winds. Figure 10(a) provides a time series of aerosol backscatter to 5 km, with a log scale for aerosol backscatter. Focusing on the MABL in Fig. 10(b), we replot Fig. 10(a) on a log vertical scale but a linear aerosol backscatter
scale. Overlaid are estimated hourly cloud bases for the lowest 10% of detected clouds from the onboard ceilometer as a standard. Given the high sensitivity of the HSRL, defining the backscatter threshold that differentiates between aerosol and cloud is needed given the system resolves the increase in backscatter through this transition. Nevertheless, the ceilometer cloud base is a fair first approximation to MABL mixed layer depth. To help with interpretation, a "Pseudo Extinction" scale is included by assuming a lidar ratio of 50 sr, the middle of the range in Fig. 4(e) and (f). In clean marine conditions where a
lidar ratio of ~20-25 sr (e.g. Fig. 4(e)) may be more appropriate, resulting in an overestimate of extinction by a factor of 2 by using this scale. Conversely, for pollution cases where a lidar ratio may reach 60 sr, the use of 50 would underestimate extinction by 20%. Thus, this depiction should be considered only for qualitative scaling purposes for biomass burning and pollution and more details will be provided in a subsequent paper on MABL lidar ratios. Using integrated aerosol backscatter, we likewise provide pseudo AOT for cloud free profiles to 7 km are provided in Fig. 10(c). Included is Pseudo extinction for
200 m altitude as a mid-mixed layer benchmark. Meteorological data from the *Sally Ride*'s bow mast is provided in Fig. 10 (d)-Precipitation, (e)-10 m temperature, sea surface skin temperature and relative humidity and (f) 10 m winds and directions. Also included are extracted (g) wind speed and (h) direction at 250, 1500, and 3000 m from the 3 hourly radiosonde releases.



Significant aerosol events were observed throughout the cruise. The largest aerosol backscatter values are overwhelmingly within MABL's mixed layer, and peaking at and around cloud base. The line plot of integrated and 250 m backscatter/pseudo extinction (Fig. 10(c)), estimated 532 nm maximum extinction coefficients were associated with major smoke events on the order of 0.3 to 0.5 km$^{-1}$ with RH<85%, nominally a visibility of only 6 to 4 km, respectively, associated with the Koschmieder coefficient of 1.9 (Griffing, 1980; Husar et al. 2000). These are similar to the P-3-HSRL2 for smoke flights. Corresponding 532 nm Pseudo AOT for cloud free paths to 7 km for events range from 0.3-0.6. The 1 Mm$^{-1}$ sr$^{-1}$ line, indicated by the blue to green transition in the colorbars, is a good qualitative indicator of polluted conditions. This level is at approximately 1-2 km, with isolated events at 2 to 3 km and the highest notable plumes detected on Sept 17 to the 0$^\circ$ C level at ~5 km. This was confirmed with personal observations on the ship. Again, given a choice of 50 sr as a lidar ratio, this estimated extinction and AOT is probably underestimated by ~0.05-0.1 km$^{-1}$ and 0.06-0.12, respectively. Conversely, pseudo extinctions and AOTs were as low as 0.1 km$^{-1}$ and 0.10, but again the true values were likely half these values. Thus, overall, we expect estimated AOTs and extinction coefficients to be under-dispersive.

Estimated MABL cloud base heights throughout the cruise ranged from a low of 300 m to a high of 800 m, with a typical value of ~450 m during the SWM, rising to ~650 m during the NWM. Examination of Fig. 10(b) and as discussed further on cloud edges are frequently observed to be associated with an overall enhanced aerosol backscatter to 1500-3000 m, with occasional altocumulus layers on top.

Examining the different panels of Fig. 10 as a whole, one can see significant aerosol and meteorology variability at both low and high frequencies. Aerosol backscatter in Fig. 10(b) has a distinct striping nature well beyond what can be attributed to precipitation or very high relative humidity, with events arriving and departing quite suddenly. Even within individual aerosol events aerosol backscatter/pseudo extinction often has a ~20-30% scatter in variability. While one would think that being in the Southwest Monsoon there would be continuous rainfall, in reality precipitation is in scattered cells, leading to short deluges and occasional light stratiform precipitation (Fig. 10(d)). For indicators of monsoonal state (Fig. 10(e)-(h))-southwest monsoon also showed variations in strength, building though the cruise with an abrupt transition to a winter northeast monsoon mode on September 22. A flow reversal to easterly winds on September 10-13 was due to a tropical depression to the east bringing additional variability in aerosol backscatter. Temperature and relative humidity also tracked monsoonal strength, with slight diurnal cycles punctuated by occasional cold pools and convective development (Fig. 10(f)). The air sea temperature differences were consistently unstable; typically skin SST minus 10 meter air temperature was 5$^\circ$C, punctuated by cold pools reducing the still unstable value in magnitude to 3.5$^\circ$C. There were occasional very slightly stable periods characterized by skin SST minus air temperature values of -0.5 $^\circ$C, likely due to advection from Asia (e.g. just September 18-19, and 21 out of the total 22 day cruise time series). Comparing monsoonal state back to the lidar profiles, aerosol events track monsoon enhancements in wind. Directional vertical wind shear is also at times noticeable. This leads to the aforementioned shearing phenomena that reinforces a low-level transport pattern.





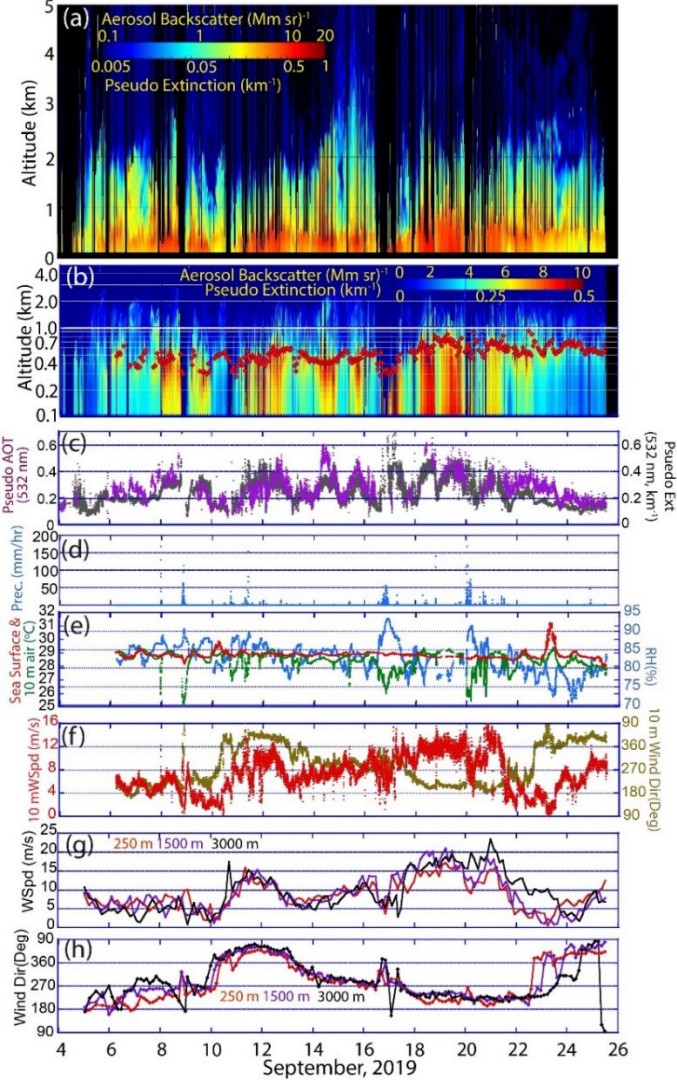

**Figure 10. September 2019 timeseries of UW-HSRL aerosol backscatter and surface meteorology from the *R/V Sally Ride* bow mast. Meteorology data began when the *Sally Ride* arrived on station on 6 September, 2019, (a) Cloud screened aerosol backscatter at 532 nm on a log scale colorbar. Included is a "pseudo extinction" for scaling based on an assumed lidar ratio of 50 Sr. This will likely underestimate clean marine conditions by a factor of two, and underestimate extinction for fine dominated environments by ~20%. (b) same as (a) but on a log vertical scale and linear aerosol backscatter scale. Overlaid in red is estimated cloud base; (c) For reference, pseudo Aerosol Optical Depth (AOT) and extinction at 200m using an assumed lidar ratio of 50 Sr, (d) Surface rain rate; (e) 2m temperature and relative humidity and sea surface skin temperature; (f) 10 m windspeed and direction; (g) Radiosonde derived wind speed at 250, 1500, and 3000 m; (h) Radiosonde derived wind direction at 250, 1500, and 3000 m.**





**Figure 11. Marine Boundary and lower free troposphere statistics of UW-HSRL aerosol backscatter from the *R/V Sally Ride* based on 3-hour increments. (a) Average and standard deviation of 532 nm aerosol backscatter taken at 200 m (mid mixed layer) with a second scale of "pseudo extinction" assuming a lidar ratio of 50 sr$^{-1}$; (b) 50$^{th}$ and 84$^{th}$** 795 **percentile of integrated aerosol backscatter taken from cloud free samples to 5 km; (c) Percentage of integrated aerosol backscatter within the Marine Atmospheric Boundary Layer's (MABL) mixed layer taken as MABL cloud base; (d) The ratio of aerosol backscatter taken at 200 m for samples under MABL clouds and those that are cloud free.**

We summarized the observed variability in aerosol backscatter by performing a series of statistics for 3 hour increments in Fig. 11. Over the course of the deployment, cloud and precipitation free aerosol backscatter spanned almost a factor of 5 (Fig.

800 11(a)), with Western Pacific air sampled at 1.8 (Mm sr)$^{-1}$ during a monsoon break on September 10, and during the NEM on



September 24[th]. Mean values reached 9 (Mm sr)[-1] for aforementioned monsoonal enhancements, or an extinction of roughly 0.45 km[-1] if one assumes a 50 sr[-1] lidar ratio. It is notable that there is also significant variability in backscatter deviations, being relatively low in the NWM monsoon breaks and NEM, and increasing in the monsoon enhancements. As with the P-3 observations, the highest backscatter values are in the MABL, with the 50[th]% in integrated aerosol backscatter being at 400-700 m (Fig. 11(b)), and the 84[th] % a kilometer above that-reaching a high of 2 km mid mission. If we take the MABLs mixed layer to be cloud base taken as the lowest 10% of clouds from the ceilometer plus 200 m for cloud entrainment/detrainment zone, 25-60% of integrated aerosol backscatter is with the MABL's Mixed layer, averaging 50% (Fig. 11(c)). Also as noted in the P-3 analyses using the HSRL-2's 0.5-second data and in Kunkel et al. (1977) and Reid et al., (2017), aerosol backscatter enhancements exist under clouds owing to updrafts. We can in part quantify this variability by taking the ratio of aerosol backscatter at an altitude of 200 m for those UW-HSRL samples with observations attenuated in the lowest 800 m representing cloudy conditions, to those with adequate sample to 5 km representing clear (Figure 11(d)). Almost all samples show ratios above 1, with 10-20% enhancements being typical.

## 5.2 Example periods of aerosol variability

It is helpful to examine short periods of the *Sally Ride* data to better interpret the long time series presented in Section 5.1. Indeed, like performed for the P-3, here we examine the nature of MABL variability for different aerosol regimes, but with a bottom-up instead of top-down perspective. The SSEC HSRL typically had a scan pattern between vertical stare and horizontal scan, repeating every 5.5 minutes. This made isolating individual cloud features difficult. However, there were several periods of vertical stare of several hours that allow for more detailed analysis. Whereas Fig. 10 images were made with 30 second/30 m data where available, for these long periods of vertical stare, data were reprocessed for continuous 5 second, 7.5 m vertical resolution. At nominal 10 m/s wind speeds, data are functionally ~50 m in horizontal resolution, as opposed to resolution ≥1 km for the P-3 and *Sally Ride* baseline observations.

### 5.2.1 Fair weather MBL

Based on Fig. 11(a), one of the lowest aerosol backscatter periods in magnitude and variability observed on *Sally Ride* was the post monsoon transition to the NWM at the end of the cruise, September 24-25, 2019. Nominally we consider this mostly Western Pacific "fair weather marine" conditions sampled by *Sally Ride*, given its low backscatter conditions and moderate regional northeasterly winds. No precipitation was detected in and around the vessel at this time. Nevertheless, given the analysis of Hilario et al. (2021) and Section 3, some East Asian influence cannot be categorically excluded. Indeed, earlier that day, the P-3 was in the vicinity of the *Sally Ride* and a lidar ratio of ~40 sr was found in the region, indicative of a mixed environment of fine and coarse particles. Retrievals of AOT at 550 nm for Terra/Aqua MODIS and at 532 nm for the LaRC HSRL for this period were also on the order of 0.15-0.2-above what is more typical for clean marine conditions (e.g. Reid et al. 2023). Because this is our most ideal "fair weather marine" baseline case, we go into more detail on the nature of the





shipboard observations (Fig. 12). Subsequent subsections 5.2.2 and 5.2.3 cover more briefly highly wind sheared and heavy smoke environments.

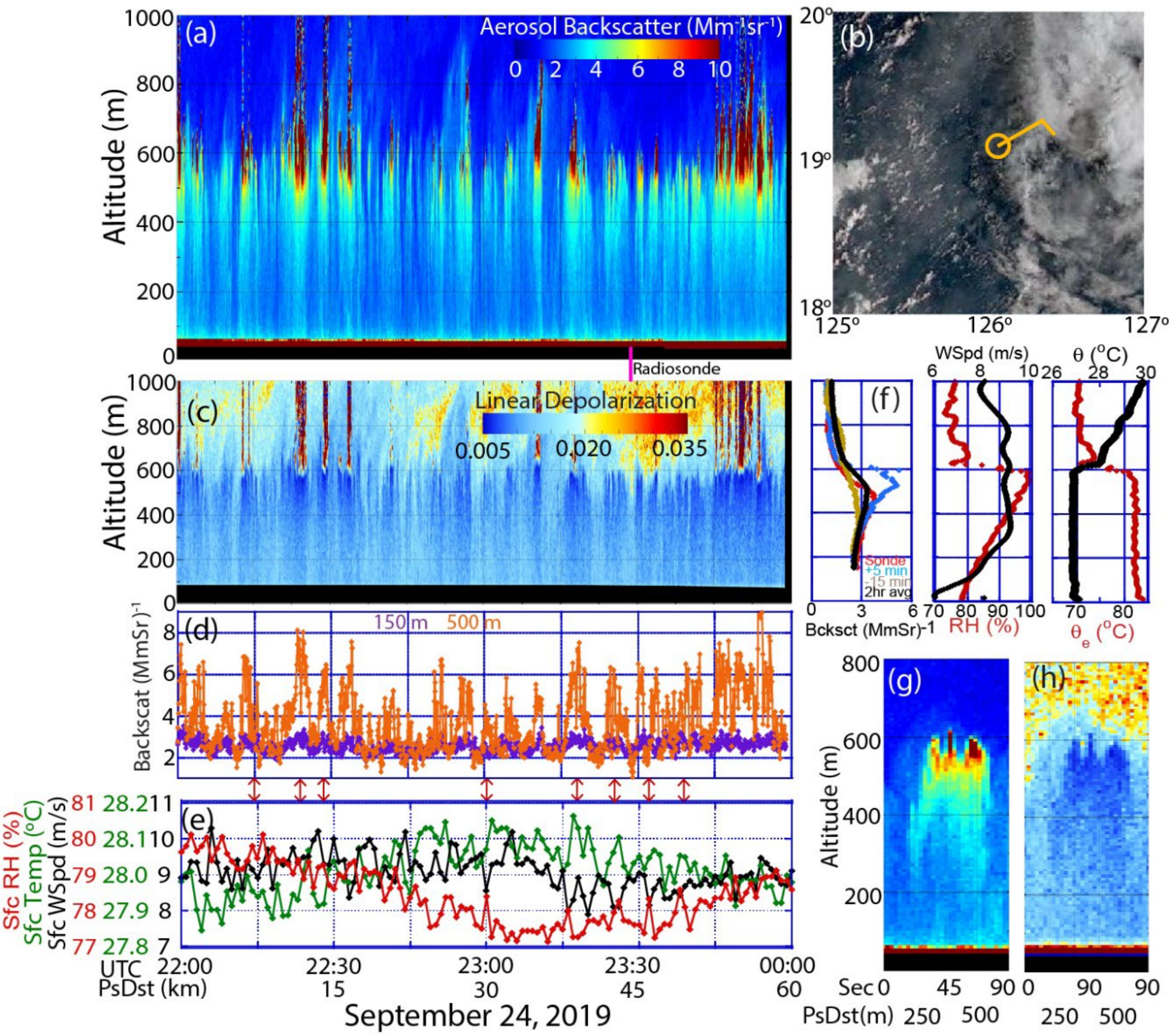

**Figure 12. A selected contiguous timeseries of lidar data on September 24, 2019 during the more maritime Northwest monsoon. While wounds are from the North Tropical Western Pacific, there is still likely some influence from Asia. Included are (a) Aerosol backscatter up to 1000 m for 22:00-0:00Z; (b) AHI imager RGB of the region, with the *Sally Ride* location and surface wind barb; (c) Linear depolarization ratio timeseries;(d) Aerosol backscatter timeseries from 150 m and 500m; (e) Time series of bow mast/surface relative humidity. temperature and winds speed on the Sally Ride; (f) At the time of the 23:27 UTC radiosonde release aerosol backscatter, and sonde relative humidity (RH), potential temperature (θ), and equivalent potential temperature ($\theta_e$);(g) and (h) Aerosol backscatter and linear depolarization (same color bar as (a) and (c)) for 90 seconds just after the radiosonde release. Marked is the radiosonde**





**release (magenta line), and notable areas of positive correlations of aerosol backscatter and temperature/anticorrelation with wind speed (red arrows).**

Figure 12 provides a synopsis of a two-hour period from September 24, 2019 22:00Z to September 25, 2019 00:00Z. Aerosol backscatter time series data (Fig. 12(a)) were collected in an area of scattered cumulus clouds adjacent to northeasterly MCS outflow (Fig. 12(b)). Notable is the frequent striping nature in aerosol backscatter below the fair Cu (red colors with backscatter >10 Mm$^{-1}$ sr$^{-1}$). Enhanced backscatter features under clouds are consistent with cloud updrafts, atmospheric roll phenomena,

and associated particle transport and hygroscopicity (Kunkel et al., 1977). Particle hygroscopic growth is further indicated in the linear depolarization (Fig. 12(c)) as more water content leads to more spherical particles relative to a refractory core with shrinking volume fraction; areas of enhanced backscatter have lower depolarization in the mixed layer. Overall, while these depolarization values in the mixed layer are quite low, it is nevertheless supportive of this being a "mixed" environment of both sea salt and a combination of Asian pollution and perhaps residual smoke from the Maritime Continent. The lowest

depolarizations are below cloud base. However, once a cloud forms, particles grow to sufficient size as cloud droplets that they depolarize again. Above cloud level, even in clear air, particles have dried somewhat and higher depolarization appears.

The striping patterns in aerosol backscatter and linear depolarization are quite vivid and can be more readily quantified in the generation of time series (d) including 150 m, the altitude at which we are most confident in near surface returns, and 500 m, 50-100 m below cloud base. As expected, higher altitudes in the ML exhibit higher backscatter. At 150 m, the amplitude in

aerosol backscatter varies between 2-3 Mm$^{-1}$ sr$^{-1}$, 20% standard deviation around the mean and at 500 m 1.6 to >8 (Mm sr)$^{-1}$, or almost a factor of two standard deviations around the mean. Notably, at 500 m values in aerosol backscatter frequently drop below the 150 m values. These oscillations are consistent with Figure 12 (d) in that they are most often associated with clouds/between clouds and extend from cloud base to the surface.

Clearly, these areas of aerosol backscatter enhancement extend to near the surface, modulated by the presence or absence of

865 clouds. Individual events are also much greater than what would be expected from the bulk analysis presented in Fig. 12(d). On this particular day, the frequency of cloud observation was 5 minutes or roughly ~2.7 km assuming a mean wind speed of ~9 m/s. At the 5-second sample level, correlations of aerosol backscatter between 150 and 500 m are only moderate (r=0.42; r$^2$=0.21), in part because of the significant nonlinear behavior below cloud base. Performing 30-second averages, correlations improve to r=0.60; or r$^2$=0.36.

While the lowest altitude we are very confident in lidar observations is 150 m, there are indications of these enhancements reaching ship level. The time series of *Sally Ride* 1-minute relative humidity, temperature and surface wind speed from the bow mast show the telltale signs of sweep and ejection events associated with variations in aerosol backscatter (Fig. 12(e)). Marked as red arrows between Fig. 12((d) and (e) r are notable isolated peaks in backscatter, and these correspond with





increases in temperature and decreases in horizontal wind-consistent with a surface ejection event (positive sensible heat flux, negative moment flux). Corresponding adjacent drops in temperatures and increases in wind speed likewise are a result of large-scale downdrafts (so-called sweeps). Humidity for this cruise was measured with an aspirated system with a response time of nearly 30 seconds, and values were subject to long period variability. Thus RH does not correlate as well as temperature and wind, but nevertheless, small amplitude peaks in RH likewise correlated with temperature and anticorrelate with wind. While a full analysis of the relationship between surface air sea fluxes and their relation to individual clouds is outside the scope of this survey paper, the preliminary evidence strongly suggests such connections between surface fluxes and individual clouds. Indeed, the strong amplitude in 500-m aerosol backscatter in Fig. 12(d) shows how structured variation exists around cloud base. This leads to questions about sampling and the distinct "modal" nature of aerosol backscatter below and between clouds.

Examining above the ML in the MABL's entrainment zone, we also see signs of inhomogeneity. There are sharp delineations in aerosol backscatter and depolarization around cloud level. Areas of particle aerosol enhancement are also decorrelated with aerosol depolarization above 600 m, a sign of particle detrainment. This said, some of the increased depolarization may be due to the extremely clean environment of the free troposphere. We can examine this further with radiosonde data. The challenge with applying radiosonde data to high-resolution lidar analysis is observations can be aliased; we cannot exactly match up the sonde with the lidar-wind shear and the evolving MABL. However, the sondes are quite useful in describing the overall environment. Marked under Fig. 12(a) is the radiosonde release time of 23:26 Z. In Fig. 12(f)we provide the aerosol backscatter at that time, +5 minutes, -15 minutes, and the 2-hour average of contiguous data, along with the radiosondes wind speed, relative humidity, potential temperature, and equivalent potential temperature profiles. While the mixed layer top is often taken as cloud base and derived as the first peak in backscatter or at some threshold value, presumably at a peak in RH, we can see in Fig. 12(a) that such definitions can be problematic. The inversion "entrainment zone" of mixed layer free troposphere exchange is somewhat ambiguous, extending well into and above the mixed layer; again noting the detraining aerosol plumes are quite visible in Fig. 12(a) and (b). Profiles in backscatter are quite variable depending on if it is taken under or between a cloud, with a 2-hour average smoothing these states together. It is also noteworthy that the lower values of depolarization extend well beyond the peaks in aerosol backscatter. Comparing these profiles to radiosonde profiles is likewise problematic, because it can take up to 90-120 seconds in a sheared environment to reach 500 m. Nevertheless, we do observe some "better behaved" profiles. RH systematically increases to 98.8% at 580 m while potential temperature and equivalent potential temperature stay roughly constant, with the exception of some near-surface instability below 40 m. At the top of the mixed layer, RH and equivalent potential temperature rapidly drop, and potential temperature rapidly increases. This is consistent with conditions supporting a moist MABL in fair weather conditions. While wind shear is visible in the lowest few hundred meters, the winds aloft are fairly uniform at 9 m/s with wind directions ranging from 40-60 degrees.



Finally, we can examine all of the data above in the context of a single cloud. Figure 12(g) and (h) show aerosol backscatter and linear depolarization around a faint cloud observed a few minutes after the radiosonde release. Clearly visible are an inner updraft core of ~250 m in diameter, of high backscatter and lower depolarization surrounded by a 100-m thick structured "halo" of related cloud influence. Notable are some areas of enhanced backscatter protruding beyond and below the cloud.

**5.2.2    Convectively active marine environment.**

We can compare the "fair weather marine conditions" with a more convectively active regime. Around September 12th, the *Sally Ride* was in relatively clear air surrounded by tropical depression Marilyn and two other MCSs. In effect, the *Sally Ride* was within the middle of the monsoon trough. Figure 13 provides similar information as Fig. 12 for 4:00-6:00Z on September 12th, 2019. During this period while being surrounded by convective systems and precipitating fair weather clouds, the *Sally*

*Ride* sampled a series of shallow convective lines advecting from depression Marilyn to the northeast. Unlike more typical roll structures that are aligned parallel to the wind shear vector (longitudinal roll clouds), these cloud features were oriented transverse to the prevailing MABL wind shear vector (transverse roll clouds). Despite the convective nature of the region, no precipitation was detected at the ship for this short investigation period. Comparing to our fair-weather case, we see significant structural differences in aerosol backscatter (Fig. 13(a)) and depolarization (Fig. 13(b)), areas of aerosol backscatter

enhancement are not evenly spaced, but in distinct areas associated with cloud line passage. Under clouds, high frequency variability still exists. As air is advecting from tropical depression Maralyn with greater wind speeds, enhanced sea salt related backscatter is likely well above the Sept 24th case. With the absence of any continental influence, depolarization is likewise greatly diminished (Fig. 13(b)), almost undetectable. Areas of aerosol detrainment between 700 and 1200 m are more strongly evident. Variability in aerosol backscatter at 405 m (50 m below nominal cloud base) relative to 150 m is far greater than the

September 24th case, but backscatter at levels throughout the ML are nevertheless correlated (Fig. 13(d)). Likewise, peaks in the anticorrelation in surface wind and temperature appear to be related to backscatter enhancements (Fig. 13(e)) although at longer temporal periods than the September 24th case as well. In this environment, correlation to RH are less dramatic.

One of the interesting aspects of the September 12th case is its greater wind speeds and vertical wind shear. The 5:29 UTC radiosonde release occurred during the passing of a line of clouds. Included in Fig. 13(f) are 5 coincident 5 second data

collections as the cloud passed over the site. The first profile at 5:29:15 UTC was just as a cloud exited, with a peak backscatter of 200 Mm$^{-1}$ Sr$^{-1}$ at 600m. Five seconds later or ~ 75 m, backscatter was down to 20 (Mm sr)$^{-1}$ reaching a minimum of 8 (Mm sr)$^{-1}$. Progressively, 15 more seconds after that with a peak in backscatter decreased in altitude and magnitude to 400 m, resolving an exchange event. Notable in the associated soundings are not only higher winds than the Sept 24 case, but more vertical wind shear - increasing from 8 m/s at the surface to 16 m/s at cloud level. Nonlinearities in the relative humidity,

potential temperature, and equivalent potential temperature plots show evidence of highly localized mixing. Indeed, if we



examine a short five-minute period of lidar data around the radiosonde release (Fig. 13(g) aerosol backscatter and (h) linear depolarization) we can clearly see the relationship between wind shear and the slanted aerosol backscatter returns as well as the presence of a detrainment layer at 700 m just above the cloud level enhancement.

Because of wind shear, the correlations between 150 m and cloud base are more difficult to quantify, although we can
qualitatively see them at larger scales. Taken at 5-second resolution, for the 2-hour period 150 and 400 m aerosol backscatter correlations are difficult to compute as cloud edge effects generate significant variability (Fig. 13(d)). But if we exclude all cases where 405 m backscatter is higher than, say, 20 Mm-1 Sr-1, correlations are still reasonable-even with shear (slope of 1.75, r=0.54; $r^2$=0.3). Performing a lag correlation results in a slight increase to an $r^2$ of 0.32 after 25 second. Averaging to 30 seconds sees significant improvement, with a slope now of 3.5 and r=0.8; $r^2$=0.66).

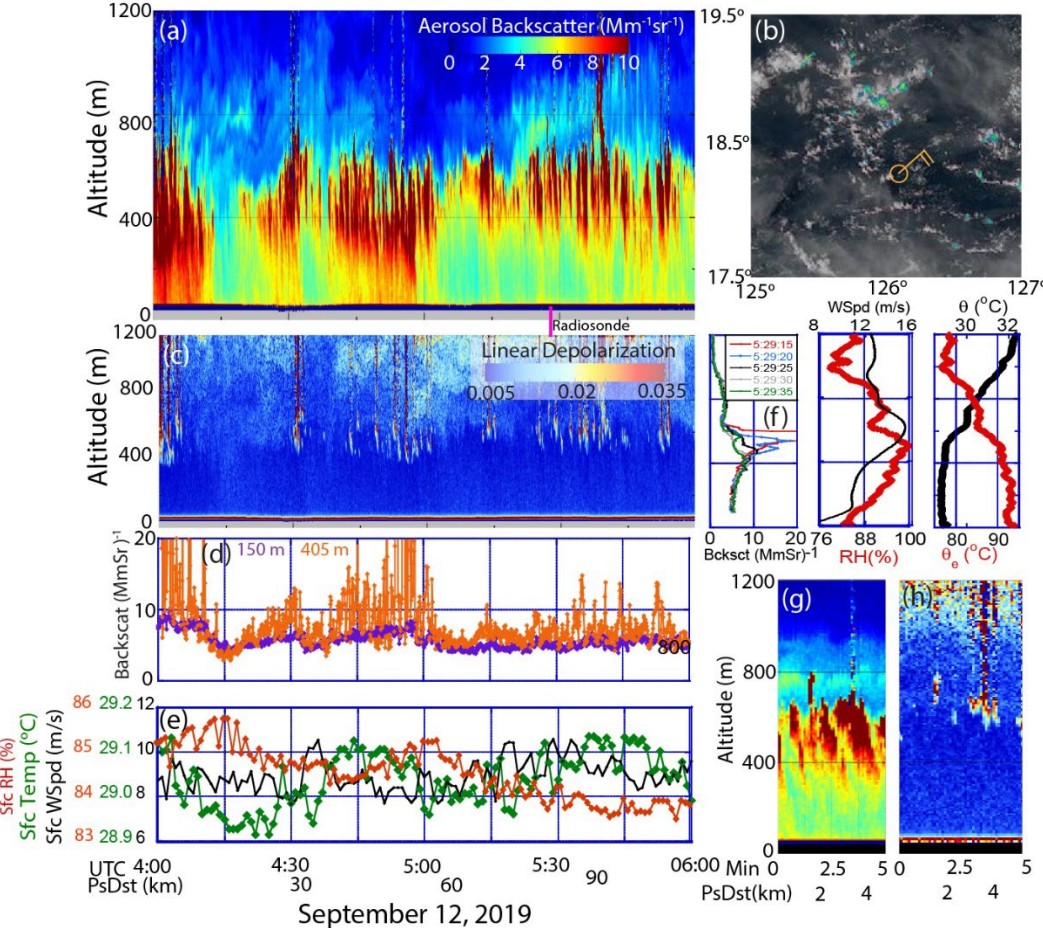


**Figure 13. Similar to Fig. (12) but for a selected contiguous timeseries of lidar data on September 12, 2019 examining outflow from a tropical system. Included are (a) Aerosol Backscatter up to 1000 m for 22:00-0:00Z; (b) AHI imager RGB of the region, with the *Sally Ride* location and surface wind barb; (c) Linear depolarization ratio timeseries;(d)**





**Aerosol backscatter timeseries for 150 m and 500m; (e) Time series of bow mast/surface relative humidity, temperature**
**and winds speed on the Sally Ride; (f) At the time of the 23:27 UTC radiosonde release aerosol backscatter, and sonde**
**relative humidity (RH), potential temperature ($\theta$), and equivalent potential temperature ($\theta_e$);(g) and (h) Aerosol**
**backscatter ad linear depolarization (same color bar as (a) and (c)) for five mintues just after the radiosonde release.**
**Marked is the radiosonde release (magenta line), and notable areas of positive correlations of aerosol backscatter and**
**temperature/anticorrelation with wind speed (red arrows).**


### 5.2.3 Heavily polluted marine

Finally, we can quickly contrast the cleaner cases of Sept 24[th] and 12[th] to the most polluted period sampled by the *Sally Ride*
during PISTON, ~September 16-18, 2019. The P-3 flew September 16 and 17, and the *Sally Ride* was forced to disengage to
the north to avoid severe weather. The early morning of September 18[th] the *Sally Ride* started to transit back to its station when
it intercepted the thick polluted air mass as it entered a tropical system (Fig. 14, similar format to Fig. 12 but note significant
scale changes). Like the more-pristine days, while we see similar features of enhanced backscatter and suppressed linear
depolarization under clouds, the situation overall is much more complex. The monsoon enhancement comes with even stronger
winds and wind shear than the previous cases reviewed. Notable detrainment layers with altocumulus formation were observed
for the last 30 minutes of the time series. Backscatter also correlates vertical within the ML but with a lesser increase with
altitude (~1.3 from 150 m to 500 m or lowest observation level to 50 below cloud base)-consistent with less hygroscopic smoke
versus more sea salt and sulfate for the cleaner marine case in Section 5.2.1 and 5.5.2-but with stronger outliers. Positive
correlations between backscatter and temperatures, and anticorrelation with wind speed exist, but also on a larger scale, in
association with what may be more organized cloud-related circulations. Perturbations to RH, if they exist, are below detectable
limits. The sounding is not as well behaved as the previous, with multiple detrainment levels visible; even variables like RH
that are typically monotonic in the mixed layer show significant variation. Mixed layer height is at 600 m, and cloud tops in
the lidar data correspond well to a second inversion at 1000 m. Indeed, for the one clearly "cloud free" case near radiosonde
release aerosol backscatter was higher aloft along with many fine detrainment layers. In nearly every sense the polluted marine
environment is much more complex; it is difficult to find an individual cloud (e.g. like Figure 13(g) and (h)). Spatial
inhomogeneity and the effect of wind shear is significant in both backscatter and linear depolarization, and even within the 2-
hour period different regimes are visible.






**Figure 14. Similar to Figure (12) but for selected contiguous timeseries of lidar data on September 18, 2019 during the most polluted period samples during PISTON. Note significant scale changes from the previous figures. Included are**

**(a) Aerosol backscatter up to 1000 m for 22:00-0:00Z; (b) AHI imager RGB of the region, with the *Sally Ride* location**





**and surface wind barb; (c) Linear depolarization ratio timeseries; (d) Aerosol backscatter timeseries for 150 m and 500m; (e) Time series of bow mast/surface relative humidity, temperature and winds speed on the Sally Ride; (f) At the time of the 23:27 UTC radiosonde release aerosol backscatter, and sonde relative humidity (RH), potential temperature (θ), and equivalent potential temperature (θₑ);(g) and (h) Aerosol backscatter and linear depolarization (same color bar as (a) and (c)) for five minutes just after the radiosonde release.**

### 5.2.4   Cold Pools

As noted in Section 4.4 transient cold pool events may be an important driver of aerosol lifting out of the MABL and into the free troposphere. Based on the *Sally Ride*'s near-surface air temperature time series in Fig. 10, six rapid temperature drops of greater than 1°C colocated with sharp near-surface wind speed increases were observed during the cruise. Of these six, two occurred while the SSEC HSRL was viewing vertically, between 16:00-23:00 Z on September 20, 2019. The first of these two, between 16:00 and 8:00Z, had superior viewing conditions and is used as an example in Fig. 15 following a similar figure format as Figs. 12 through 14. However, unlike previous figures given the dynamic range of the observed lidar signals, log scales are used.

The aerosol backscatter profiles for a ninety-minute period are provided in Fig. 14(a). At the beginning of the time series, we see similar aerosol backscatter features to the convective environment in Figs. 12(a) and 13(a). However, notable is a heavy rain shaft at 16:20Z, followed by that storm's cold pool with very clean air above 200 m at 16:30. Another minor event was observed just 15 minutes later. This cold pool occurred at night within one of Typhoon Tapa's convective arms, and only IR imagery is available (Fig. 15(b)). Like the previous active convection and clean (presumably sea salt dominated) conditions, there is virtually no linear depolarization signal associated with aerosol particles, although the rain shafts are clearly delineated around and above 0.2 resulting from multiple scattering within the precipitation. While overall, comparisons of nearer surface (e.g. 150 m) versus mixed layer top are similar to other cases, near the cold pool boundaries vertical and horizontal gradients in aerosol backscatter between the near surface and upper mixed layer environments can be quite dramatic (Fig. 15(d)).

The influence of the cold pool is well described by *Sally Ride'*s observation of a drop in air temperature and sharp increase in wind speed and RH (Fig. 13(e)). Given the presence of rain shafts next to clear air, aerosol backscatter signals span two orders of magnitude in this data segment. Ultimately, the near cold pool environment results in more varied aerosol and thermodynamic profiles compared to more benign conditions. Included in Fig 15(f) are example aerosol backscatter profiles within and outside of the cold pool near radiosonde release. Within the cold pool (16:34 and 16:35Z) aerosol backscatter profiles are essentially static in the MABL, except for a near surface aerosol backscatter enhancement. After the atmosphere had recovered, the MABL sounding and aerosol profiles were much more typical.

Magnifying on the cold pool area (Fig. 15(g) and (h)), strong enhancements of aerosol backscatter very near the surface are associated with the enhanced winds along the cold pool boundary. As noted in the P-3 case in Section 4.4 and in similar observations of aerosol particles in similar convective regimes in the South China Sea (Reid et al. 2015), these near surface backscatter enhancements may be due to wind driven sea spray-essentially sea salt haboobs.


**Figure 15. Similar to Figure (8) but for two transects across an evolving cold pool for RF 7 around 20 September, 2019 between 16:00 and 17:30Z. Included are (a) Aerosol backscatter up to 1000 m; (b) AHI thermal infrared image of the region, with the *Sally Ride* location and surface wind barb; (c) Linear depolarization ratio timeseries; (d) Aerosol**



**backscatter timeseries for 150 m and 450m; (e) Time series of bow mast/surface relative humidity, temperature and**
**winds speed on the Sally Ride; (f) At the time of the 17:29 UTC radiosonde release aerosol backscatter, and sonde**
**relative humidity (RH), potential temperature (θ), and equivalent potential temperature (θe);(g) aerosol backscatter**
**(same color bar as (a)) for 10 minutes after cold pool passage starting at 16:30Z.**

## 6.0 Conclusions and Implications

The purpose of this paper is to outline the modes of variability in aerosol vertical profiles over the North Western Tropical
Pacific (NWTP) using the combined assets of the ONR Propagation of Intraseasonal Oscillations (PISTON) R/V *Sally Ride*
stationed East of Luzon and the NASA Cloud, Aerosol and Monsoon Processes Philippines Experiment (CAMP²Ex) P-3 flying
over the Sulu, South China and Philippine Seas. These cooperative experiments were deployed in the late boreal summer of
2019, and fortuitously captured the monsoonal transition from the large-scale Southwest to Northeast flow. Consequently, the
platforms observed a wide variety of aerosol sources, including significant biomass burning activity and pollution emissions
from the Maritime Continent (MC), Asian pollution, and pristine air from the NWTP and convective outflow. The focus in the
paper is on both the P-3's and *Sally Ride*'s High Spectral Resolution Lidars (HSRLs) to outlines the "animals in the zoo" of
aerosol vertical phenomenology as an introduction to the rationale for needed efforts in understanding how aerosol
inhomogeneity relates to the heterogeneous nature of aerosol transport/exchange/transformation, radiation, and flux/exchange
fields. Throughout, we note areas of ongoing work that reference this overview paper.

As has been noted in aforementioned work (e.g. Hilario et al. 2021; Edwards et al. 2022; Reid et al. 2023; Xiao et al. 2023)
and presented in more detail here, P-3 in situ and *Sally Ride* HSRL observations highlight the MC's distinct monsoon driven
lower free troposphere and long-range transport phenomenology. As a consequence, the larger NWTP's Marine Atmospheric
Boundary Layer (MABL) and lower free troposphere are impacted by a very wide range of aerosol properties. Unlike most
subcontinental to continental scale aerosol events where long-range transport is facilitated by the formation of land plumes
and advection by free tropospheric winds, the 2019 PISTON and CAMP²Ex missions verified the dominance of MABL and
lower free troposphere aerosol transport pathways of thousands of kilometers with 532 nm Aerosol Optical Thickness (AOT)
at times above 1. Background marine, Indonesian biomass burning outbreaks, MC emissions and Asian pollution cases were
observed to have 95% of integrated aerosol backscatter below 3 km in altitude in both P-3 and *Sally Ride* observations. Twenty
to fifty percent of total aerosol backscatter and extinction are contributed by the MABL's 400-650 m high Mixed Layer (ML),
with the remainder mostly falling off with altitude in the free troposphere, with intermittent cloud detrainment layers.
Conversely, aerosol backscatter is quite low in the middle and upper free troposphere. All three cases of P-3 observations near
the *Sally Ride* point to good comparisons of aerosol backscatter profiles between the instruments in the MABL overall, but
demonstrate significant variability in spatial scale of aerosol layering in the lower free troposphere. These different "top down"



versus "bottom up" views agree in the nature of lower-level transport, but there are noticeable differences in the benchmark calculation of fraction of aerosol backscatter that is in the boundary layer. Closure studies are underway.

There are several key physics findings in the analysis that are related to scale, from the large-scale monsoon to the role of convection and the importance of individual updrafts.

1) *Monsoonal Maintenance*: In this paper, we expanded on Edwards et al. (2022) analysis of the Navy Aerosol Analysis and Prediction System Reanalysis (NAAPS-RA) simulations. At the largest monsoonal scale, what maintains the primary low-level aerosol transport process is a combination of low injection heights of smoke and pollution off of the MC islands (as noted in Wang et al., 2013) coupled with veering directional wind shear with height. Thus, particle transport naturally bifurcates in altitude, with the near surface particles remaining in and around the MABL during long range transport in the SWM flow. Any

aerosol layers aloft from either enhanced injection or convective pumping would overall be preferentially advected more eastward than those in the MABL. As demonstrated in Bukowski et al. (2019), above the melting level (~4.8 km) monsoonal flow reverses direction completely to northeasterly. The P-3 observations clearly observed a reduction in aerosol layering aloft between just offshore of Borneo and 1000 km to the northeast. The absence of strong aerosol layering above 3 km suggests a combination of low level free tropospheric inversions capping sometimes active warm convection and efficient wet scavenging

as contributors to keeping aerosol layering heights low. Further, even though the region is convectively active, it is only through major tropical features or significant coastally forced convection that major scavenging takes place. Consequently, fine mode particle lifetimes in or just above the MABL can be extraordinarily long lived in this SWM system lasting over 4000 km. This work supports the long-standing suggestions by Clarke et al. (1996; 2013) and Quinn et al., (2017) that note that even in the most remote marine areas, cloud condensation nuclei populations can be heavily influenced by terrestrial sources. However,

while Clark et al.'s observations suggest mixing from the free troposphere into the MABL, this present work shows that particle can be exceptionally long lived in the MABL only to be scavenged in significant storm features. Because these primary maintenance mechanisms are at the large scale, the global models replicate this long-range transport quite well as shown in the NAAPS-RA. The extent that smoke as CCN may inhibit precipitation and thus *significantly reduce* scavenging is still an open question for CAMP$^2$Ex,

2) *Lower free troposphere layering:* While the large-scale flow helps sustain long-range low-level aerosol transport, in our discussions with other regional scientists investigators are still at times surprised by the dominance of the MABL and its mixed layer over the lower free troposphere for the highest particle concentrations and aerosol backscatter. With so much convection, it is argued that more detrainment above the MABL and scavenging within the MABL should be expected. Indeed, only once in this analysis (the P-3 RF6 case on September 6th, Figure 5 (h)) did we find a situation of maximum particle concentrations

above the MABL. But even here it might be explainable again as a combination of vertical wind shear to the east of an area of active warm convection capped by a strong mid-level inversion and potential cold pool lofting. However, it is noteworthy that while the highest concentrations and maximum aerosol backscatter are in the shallow MABL, the integral of particle mass





species and backscatter is still often on par with the lower free troposphere, punctuated by frequently observed and remarkably thin (<100 m) aerosol layers, demonstrating significant convective detrainment. In cases adjacent to detraining clouds, detrainment layers are also associated with altocumulus. Further, both the P-3 and *Sally Ride* HSRL observations show how intricate the detrainment process is in and around cumulus clouds. This likely explains some deficiencies in sub-grid parameterizations in describing lower free troposphere aerosol loading as identified in Edwards et al. (2022). The absence of strong aerosol layers in the middle to upper free troposphere coupled with enhanced CO is suggestive of efficient wet scavenging for those cells that puncture lower level inversions. Although as Xiao et al. (2023) discusses, there is significant nucleation potential of detrained precursors aloft.

3) *MABL Exchange:* One of the most striking findings of this survey is how variable aerosol backscatter, depolarization, and presumably light extinction are in the MABL and its associated mixed layer. Local enhancements in aerosol backscatter of 30-100% were found underneath clouds. Just as observed by the ground observations by Kunkel et al., (1977) and the SSEC-HSRL deployment at Huntsville AL for the terrestrial NASA Study of Emissions and Atmospheric Composition, Clouds and Climate by Regional Surveys campaign (SEAC⁴RS; Reid et al. 2017), there are persistent enhancements in aerosol backscatter and decreases in linear depolarization due to moist updrafts under clouds. Cloud halos relative to large detrainment layers are also clear. The difference for the marine environment measured in PISTON/CAMP²Ex versus terrestrial observations is shallowness of the boundary layer inversion, ~450-650 m over water versus 1.5 to 2 km over land. Consequently, these under-cloud backscatter enhancements reach the ocean surface, even visibly correlated to surface winds and temperature, and presumably fluxes. High temporal resolution observations from the *Sally Ride* clearly demonstrate the formation of cloud halos and air mass exchange around clouds. Such features are also observed in P-3 HSRL-2 0.5-second data at the edges of clouds, although with the aircraft flying at 150 m/s fine features are washed out. Mixed layer backscatter enhancements are more prevalent for higher large-scale MABL wind speeds (>7 m/s) where MABL roll features and associated cloud streets form. Such features are diminished for lower wind speeds or stable conditions-which unfortunately existed during all three periods of P-3-*Sally Ride* joint observations. As expected for periods of higher MABL wind shear, aerosol backscatter enhancement features are likewise noticeably sheared (e.g. Fig. 11(g)). Nevertheless, the lidar observations clearly link cloud base to the air-sea interface. Increased aerosol backscatter is locally related to reduced wind speed and increased temperature associated with an ejection event, or ejection; decreased backscatter is associated with decreased temperature and increased wind (i.e., sweep event). Further studies are needed to establish the extent these backscatter enhancements are associated with surface fluxes, MABL/free troposphere exchange, and moisture convergence.

4) *Observational signal to noise in the HSRL:* The great advantage of the HSRL systems over other lidars is its ability to very accurately derive calibrated aerosol backscatter both day and night from the ratio of total to molecular backscatter. Consequently, even in the most pristine conditions, both HSRLs had excellent signal to noise performance yielding profiles for aerosol backscatter signals with precisions that are a small fraction of the Rayleigh signal. Error propagation is also much




more localized than more common elastic backscatter lidar systems. It is for these reasons that there are active proponents of HSRL development for satellites, as evidenced with the recent EarthCARE launch. However, extinction and lidar ratio are calculated from the gradient in the molecular signal and are thus much more sensitive to noise. This is quite apparent in Fig. 4 when comparing backscatter versus extinction products for typically marine to polluted conditions. The 10-second reported extinction products are oversampled in a boxcar 60-second window leading to functionally a 10-12 km product (versus 1.75-km 10-second backscatter product), and in the vertical the 15-m reported product is itself computed over a 300 m vertical window, that we in addition performed additional averaging. In the context of large scale polluted conditions, the HSRL provides a valuable degree of freedom in the derivation of lidar ratio. However, even with longer integration times and lengths, extinction profiles for typical conditions are nevertheless quite noisy as are associated lidar ratios for typical marine conditions and key exchange features are washed out. For the highest 532 nm AOTs measured for the clean marine conditions, RF15a on the western side of convection, lidar ratios in the 20-25 sr$^{-1}$ range are consistent with expected coarse mode, spherical sea-salt droplet dominated conditions (Burton et al. 2012), lending confidence to these measurements. Nevertheless, the lidar extinction profile, while still showing a mixed layer dominance, does not show the clean transitions as observed in the backscatter due to the coarser resolution and higher noise profile for the more tenuous aerosol. For other cases where aerosol backscatter in the MABL is a half (e.g. RF15 2) to a tenth (RF19 a) of the polluted cases, extinction profiles and lidar ratios are much more noisy and difficult to interpret-although the lidar ratio is still overall close to marine environments at 30 sr. Although aloft lidar ratios steadily rise, perhaps due to a lower prevalence of sea salt and higher prevalence of a fine mode (e.g. sea salt may have been preferentially scavenged/longer range transport of a fine mode/particle nucleation) or as a low extinction related artifact. Applying even a mean lidar ratio of 30 sr to a cleaner aerosol backscatter profile, 532 nm AOTs are much lower than what the extinction retrieval would suggest, down as low as 0.03 for RF19-1. What this suggests is that for cleaner or even more typical conditions and fine features, algorithm development should be more heavily weighted on projection of lidar ratios onto aerosol backscatter than specifically the extinction products more directly.

5) *3D variability and radiative effects:* The combined PISTON/CAMP$^2$Ex observations are spawning studies focused on the features and physical processes related to the scales outlined above. Remote sensing will no doubt play a prominent role in quantifying such processes. Historically, cloud and aerosol populations are measured in a 2-dimensional product (e.g., cloud top heights, AOT, etc). With the recent advancement of higher-level technologies (e.g. polarimetry; hyperspectral; HSRL in space, better geostationary, etc), additional complexity will no doubt be able to be monitored. However, PISTON/CAMP$^2$Ex demonstrated that just because aerosol transport is near the surface it is not necessarily simple. In fact, the opposite is true- the fine scales associated with near surface aerosol and cloud features have challenges beyond large mid and upper-level clouds and plumes. Despite the prevalence of attempts to correlate AOT with boundary layer aerosol mass or CCN populations, skill in such a regression or machine learning is highly localized. We find that as expected, use of AOT as an indicator of particle concentrations in the MABL or cloud detrainment is problematic. Despite the highest concentrations of particles being in the MABL, just as in Reid et al. (2017) over land, the variability in AOT contributions above the MABL is what often dominates





AOT variability (e.g. Fig. 10(b)). Low AOTs can still have extraordinarily high aerosol backscatter in the MABL and vice versa. Further, backscatter enhancements under clouds add a potential 3-dimensional effect for non-nadir optical paths,

becoming even more complex for high wind shear in and around the MABL. All of this points to potential challenges of extracting aerosol information in and around clouds. Indeed, as pointed out in Marshak et al. (2021) there are significant radiance enhancements due to added radiative effects-notably cloud reflections. Spencer et al. (2019) showed AOT enhancements immediately adjacent to clouds, but Cloud Physics Lidar data they used was unable to resolve or penetrate under or immediately adjacent to clouds. It requires further study to determine under what circumstance uncertainties may be larger

for any slant path application such as polarimetry or if the complexity of the system simply cancels out. Similar morphological considerations also need to be considered for clouds. A review of downward video with examples in Figs. 5 and 6 suggest aerosol detrainment is often associated with altocumulus clouds. These vertically thin but horizontally expansive clouds may be an appreciable fraction of what is perceived by satellites as cumulus clouds and are also difficult to replicate in all but the highest resolution LES models.

6) *Higher order inversions and hygroscopicity:* Questions regarding 3D radiative transfer immediately lead to questions of higher order retrievals, such as size, index of refraction and hygroscopicity. Particle hygroscopicity is a particularly important parameter to understand as water uptake in high relative humidity environments like the MABL is what largely modulates size, refractive index, and ultimately scattering. Notable from the onset of analysis of P-3 data was the finding of negative hygroscopicity within the in-situ instrumentation. The second question is to what extent we can estimate aerosol hygroscopicity

from lidar profiles. Indeed, it has been previously hypothesized by Feingold and Morley (2003) that by assuming saturation at cloud base and an adiabatic lapse rate that then aerosol hygroscopicity can be inverted. More formal theoretical investigations on extraction of particle microphysics from lidar and polarimetry are also optimistic (Kahnert and Kanngießer, 2024). However, the authors here have long experimented with the SEAC⁴RS ground SSEC HSRL and CAMP²Ex P-3 HSRL-2 data for such an extraction and papers are forthcoming. To be sure, something can be said about particle hygroscopicity and other

particle properties. Overall, RH increases systematically as expected in the mixed layer. Particle backscatter increases and linear depolarization decreases under clouds. Clearly, the particles observed in Fig. 6 and 13 are hygroscopic. However, there are several considerations to make an inference to quantitative data. First, the lidars and other remote sensors observe the ensemble of aerosol particles regardless of size. In a marine environment, there is always a mixture of particles with hygroscopic sea salt that must be controlled. Second, dry hygroscopic particles grow rapidly with RH. However, once hydrated,

particles retain their moisture as they dry, known as hysteresis (e.g. Carrico et al. 2003). This "upper" hygroscopic curve is more uncertain, and shows less sensitivity to RH. In and around the MABL, it is often assumed that particles represent the upper part of the hygroscopic hysteresis curve. However, this is not always a good assumption; Ferrare et al. (2023) has noted that depolarizing sea salt is sometimes found around MABL in areas of dry air aloft. As particles never fully dry out (RH even above the MABL is 75% as opposed to ~ 35% often required for efflorescence) we are not currently in a position to compare

quantitatively with the airborne measurements. Further, as particles grow and shrink, their lidar ratio should also change.



Finally, we know from our findings that there is heterogeneity in the MABL itself and assuming a simple adiabatic lapse rate with altitude and uniform moisture might also be problematic. While as a whole the balloon sounding is "well-behaved" and depicts the MABL as we expect, it is nevertheless a single profile in an inhomogeneous and sheared environment. We do not know specifically if the sounding is predominately in an updraft or downdraft region, or even the extent humidity changes

both vertically and horizontally  especially between clouds when airborne or spaceborne lidar observations are made. But, we do know below cloud base for even sea salt dominated environments that aerosol backscatter between clouds can be larger at 200 m than just below cloud base (e.g. Fig. 11 through 13). In comparisons, "bottom up" observations from ship samples in an Eulerian fashion suggest that small changes in distance or direction can result in differences in environments (Park et al. 2022). What is perhaps most uncertain, however, is the nature of hygroscopic growth, the hysteresis for these particles, and

associated changes in lidar ratio that must be parsed within the ensemble of particles in the beam.

## 7.0 Data availability

The PISTON *Sally Ride* dataset can be found at the NASA Langley Atmospheric Data Center at https://doi.org/10.5067/SUBORBITAL/PISTON2019-ONR-NOAA/RVSALLYRIDE/DATA001  or at the NASA Langley Airborne Science Data Repository (https://www-air.larc.nasa.gov/cgi-bin/ArcView/camp2ex?RV-SALLY-RIDE=1).

PISTON context can be gleaned from the mission site: https://onrpiston.colostate.edu/ . *Sally Ride* HSRL data can be visualized at: https://hsrl.ssec.wisc.edu/by_site/29/all/. The CAMP²Ex data doi is https://doi.org/10.5067/Suborbital/CAMP2EX2018/DATA001 with access available at https://www-air.larc.nasa.gov/missions/camp2ex/index.html. Additional descriptive and satellite data can be found on the University of Wisconsin Geoworldview site (http://geoworldview.ssec.wisc.edu/).

## 8.0 Acknowledgements

CAMP²Ex could only be performed with the persistent effort and the cooperation of many Philippine entities including the Manila Observatory, PAGASA, the Department of Foreign Affairs (DFA), Philippine Department of Science and Technology (DOST), Civil Aviation Authorities of the Philippines (CAAP), and Armed Forces Philippines (AFP). We are also grateful to the U.S. Embassy in Manila Economics and Science Department for shepherding the CAMP²Ex mission over many years of

preparation. For PISTON, we thank collaborators in Taiwan, particularly Prof. Sen Jan and colleagues at National Taiwan University, and in Palau. The PISTON and CAMP²Ex deployments were an enormous logistical undertaking.  Team members are most appreciative to the University of California San Diego Crew of *Sally Ride*, the NASA Wallops flight facility, and the NASA Earth Science Project Office.

## 9.0 Financial Support



Support for the PISTON cruise and lidar observation program was provided by the Office of Naval Research Code 322 whereas the CAMP2Ex program was supported by NASA with supplemental flight hours supported by the NRL Base program. JSR was supported by the NASA ACCMAP and ONR PISTON programs. Support for the HSRL-2 team was provided by the NASA Radiation Sciences Program.  AS was supported by NASA grant no. 80NSSC18K0148 and

ONR grant no. N00014-21-1-2115. CGA was supported by Cooperative Agreement 80MSFC22M0001 between NASA Marshall Space Flight Center and The University of Alabama in Huntsville, and CGA acknowledges additional support from an appointment to the NASA Postdoctoral Program at NASA Marshall Space Flight Center, administered by Oak Ridge Associated Universities under contract with NASA, through contract 80HQTR21CA005. SCV was supported by the Office of Naval Research Grant N00014-16-1-3093 and by NASA Grant 80NSSC18K0149. JW acknowledges support from NASA

Radiation Sciences Program (grant no. 80NSSC19K0618). EJT was supported by the NOAA Physical Sciences Lab and their shipboard surface and ceilometer data was collected and made possible by NOAA Climate Program Office in part by Award #NA17OAR432010.

## 10.0 Author contributions.

Author JSR conceived the study, performed the bulk analysis, and was lead writer of this paper. Authors JIR and PX  conducted NAAPS analysis, data processing and aided in the shear analysis. Authors CH, RF, and SBP participated in the field experiment (HSRL-2 team) and contributed to reviewing and editing of this paper. Authors REH, EWE, WJT, and RK developed the HSRL technology, calibrated/deployed/processed the SSECHSRL measurements for the deployment. Author EJT contributed to reviewing and editing of this paper, led data archiving of all PISTON datasets, and led data collection and processing of the

shipboard near-surface oceanic and meteorological, air-sea flux, and ceilometer data Authors HBM, DPE, and JIC provided financial and science support.  Author SCV participated in the field experiment (PI for dropsondes; P-3 flight scientist), was a CAMP2Ex white paper co-author, and contributed to reviewing and editing of this paper. Other coauthors that provided data and editing include  CGA (AMPR), GSD & JPD (Trace Gas) AS(AMS), ,KLT (State),  JW (FIMS),  JDZ (LARGE)

**11.0 Competing Interests**

One of the (co-)authors is a member of the editorial board of Atmospheric Chemistry and Physics."

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
