# Peer review of "PISTON and CAMP2Ex observations of the fundamental modes of aerosol vertical variability in the Northwest Tropical Pacific and Maritime Continent's Monsoon"

_EGUsphere, 2025_

## Referee Comment (RC2)

This paper documents aerosol data and observations from the 2019 ONR PISTON cruise and NASA CAMP2Ex flights performed over the north-western part of the Maritime continent on South-East Asia. The manuscript is rather large and highly descriptive of the multiple aerosol processes going on the Northwest Tropical Pacific's (NWTP) monsoon environment when multiple meteorological conditions are present during the observation period. The manuscript does an excellent job in advancing our current understanding of the interactions between meteorology, aerosols, and clouds on clean and polluted environments from a double perspective provided by aircraft and cruise measurements. This publication indeed does move the research frontier forward by providing a highly detailed description not only of the data itself but of the physics behind these multiple processes observed. Vertical and long-range transport of aerosol species and its presence over long distances and periods of time has been one of the areas for which very little measurements exist or none, let alone knowledge of its impact and implications for the Maritime continent regional environment. In addition, sections 4.4 and 5.2.4 relating the role of cold pools to particle lofting is, in my view, the biggest take home message and a very significant contribution to our understanding of aerosol lofting across this region. In essence, this work provides a fundamental baseline for future observations over the maritime region. I whole hearty recommend this article to be published and I would suggest to the authors to make a full review of the paper and identify areas to be condensed if possible.

A couple of issues to note as shown below.

Line 550: "We surmise that these features..." I would guess the authors means "We assume ..." or something along those lines.

Line 564: Table 2 is missing from the manuscript (there are other places in which Table 2 is mentioned)

Line 590: "We initially surmised that the smoke..." Same typo as in Line 550 above.

Line 940: Closing parenthesis is missing at the end of the paragraph.

---

## Author Comment (AC1)

Citation: https://doi.org/10.5194/egusphere-2025-2605-RC1

This paper contains a lot of information that will be useful for researchers applying further analysis. The manuscript did read more like a Measurement Report than a scientific analysis aiming to move the research frontier forward, and I would suggest that the manuscript be reclassified as a Measurement Report. From a science point of view, I would have wished for more assessment of why the aerosol remains as low in the atmosphere as the authors found, and, how common that is. I did not see much discussion of other campaigns, including those associated with NASA, that also have documented long-range boundary layer aerosol transport. NASA ORACLES/DOE LASIC certainly did (see ACP/AMT special issue on the southeast Atlantic), and perhaps NASA SEAC4RS did too. I had trouble taking away what new scientific insights were gleaned. But just reframing the document as a Measurement Report would remove this additional weight from the authors and that would be fine as the collected data should be published to better familiarize the larger world with it. I would also suggest the authors read through the manuscript and look for ways to condense the language, as at 50 pages for the main text, it is quite long.

Thank you for taking the time to review the paper. We recognize this one was a beast with many complex interdependencies. After much discussion among the coauthors about whether we should break it up into separate papers, we concluded that, given the interdependencies, it is better to take the approach we are currently using: one overview paper, followed by more concise papers on specific topics that can reference the system as a whole. Given how the community is advancing to tackle more and more complex systems (as was CAMP2Ex's goal), this may be open of the more efficient ways to proceed. If we provided less data and discussion, we could not support our findings. Any more and it makes an already long paper even longer. Breaking the paper up in the beginning in parts is also problematic because of the strong interdependence. So please pardon this format and accommodation.

In your major comments, you bring up several other important topics:

a) This paper read more like a measurement report: When we first read this comment, our immediate reaction was to consider that reclassification might be warranted. However, upon further consideration and after reviewing relevant literature, we believe the initial classification as a scientific paper remains appropriate. The paper references a significant portion of the CAMP2Ex and PISTON datasets, integrating them in a way that allows for the simultaneous application to different aspects of the monsoonal aerosol system. Indeed, the complexity and interrelated features of this system are what compelled us to write this overview paper as a springboard for further, more focused work. Furthermore, throughout the paper, we have striven to link the observations to specific physical processes (with extensive citations), and the final conclusions/implications section highlights the possibility that, as a community, we may need to revise our interpretation of boundary layer aerosol observations, especially with regard to lidar observations. These aspects

clearly distinguish the paper from a typical measurement report. Regarding the classification, and given the contrary opinion of reviwer two would welcome the ACP editors opinion if a change is necessary.

b) More of an assessment of why the aerosol remains as low in the atmosphere as the authors found.

This is precisely the conundrum we face. We could have explored these aspects in greater depth, but that would have made an already lengthy paper even longer. Thus, we reiterate our strategy of providing an overview paper, to be followed by more detailed analyses in subsequent publications. Regardless, we respectfully disagree with Reviewer 1's assertion that we do not adequately attribute the reasons for low-level transport or that the takeaway points are not clearly articulated. The 250 word abstract and the paper outlines the two dominant modes of variability: 1) Low-level veering vertical wind shear in the monsoonal flow, which was identified in the NAAPS reanalysis and demonstrated in the lidar data. Regarding why the source does not inject further into the atmosphere, we cite the 7SEAS work extensively throughout the paper (see bibliography below). 2) Convective elements, including cloud detrainment, cold pool lofting, and, perhaps more importantly, recirculation. We also refer to CAMP2Ex papers that demonstrate the clouds' high efficiency in aerosol particle scavenging, which results in diminished detrainment aloft (e.g., Xiao et al., 2023; Hilario et al., 2025). Nevertheless, we do provide examples of cloud detrainment into layers, often in association with altocumulus, and emphasize the importance of observing such features in relation to the up- and down-shear directions

c) Not much discussion of other campaigns, with notes on ORACLES/DOE LASIC and their low level transport.

A more detailed discussion of other campaigns, notably ORACLES, was included in an earlier draft of the paper, adding 1.5 journal pages of text. This was subsequently removed in favor of a planned paper on the nature of offshore flow. We have reinserted several sentences specifically addressing these missions in the first paragraph of the introduction. However, regarding low-level transport specifically, we maintain our assessment that severe Maritime Continent outbreaks are distinct from other subcontinental smoke transport events. Indeed, the ORACLES papers indicate that the vast majority of smoke transport occurs in the free troposphere (e.g., Mallet et al. 2019)—a well-known phenomenon dating back to the 1990s (e.g., Anderson et al., 1996; Swap et al.). A similar observation can be made regarding the associated LASIC campaign at Ascension Island; we note that Dobracki et al. was also recently published, and we cite this work. Although the mass concentrations observed in the MABL by LASIC are, on average, an order of magnitude less than those observed by CAMP2Ex, CAMP2Ex's peak mass concentration in the MABL was a factor of 40 higher than LASIC's, even after several thousand kilometers of additional transport distance. Indeed, the very definition of LASIC is "Layered Atlantic Smoke Interactions With Clouds," implying that while boundary layer smoke is present, even significant at times, it remains a

secondary transport mechanism alongside a strong free tropospheric plume structure and entrainment from aloft. Similar observations are made for CLARIFY (e.g., Haywood et al.). Atlantic smoke transport is fundamentally different from the Maritime Continent monsoonal system in terms of emissions, offshore flow, shear, and the very nature of cloud elements (e.g. stratocu versus monsoonal). We also note that observations at Ascension Island are 3000 km from the coast of Africa—a considerable distance-but CAMP2Ex measured even more significant aerosol loading at twice that distance. Nevertheless, we are keenly aware of, and cite, the importance of long-lived boundary layer aerosol transport, particularly referencing the pioneering work of Tony Clarke and later Patricia Quinn, who point out that a "true background marine" environment is hard to identify.

**d) Condensing language.**

The paper is certainly long, but it is not what we would consider overly verbose. We have tried to balance the volume of material with readability.

**Smaller comments:**

1. URLs sprinkled throughout the manuscript belong in the data availability section I believe, not in the main text.

Thanks, yes, it is always a matter of preference. We have done this before, and then have reviewers say "Oh, it would be nice to know where this is when you introduce this." But in response we have moved all http calls to the data availability section.

2. A larger map outlining the campaign location would be nice. Not everyone knows where Luzon or Manila are, or the various Seas.

We labeled Figure 2(c) with geographic information.

3. The Hilario et al paper is referenced so often that it would be nice to see it briefly summarized in the Intro. I presume it did not include any HSRL analysis.

Response: Correct, the Hilario analysis did not include any HSRL data. We added a few sentences to the introduction. As noted in the existing paper, we only needed to use this analysis qualitatively, and additional analyses are underway to deal more directly on the influence of shear and its representation in models in defining the aerosol vertical profile. Given the length of the paper already, we do not believe a more extensive description of the Hilario measurement report that what we just inserted is warranted.

**References**

**Long lived aerosol transport:**

Clarke, A. D.; Freitag, S.; Simpson, R. M. C.; Hudson, J. G.; Howell, S. G.; Brekhovskikh, V. L.; Campos, T.; Kapustin, V. N.; and Zhou, J. Free troposphere as a major source of CCN for the equatorial pacific boundary layer: long-range transport and teleconnections, Atmos. Chem. Phys. 2013, 13, 7511–7529, https://doi.org/10.5194/acp-13-7511-2013, 2013.

Clarke, A. D.; Kapustin, V. N. The Shoreline Environment Aerosol Study (SEAS): A context for marine aerosol measurements influenced by a coastal environment and long-range transport, Journal of Atmospheric and Oceanic Technology, 2003, 20, 1351-1361.

Clarke, A. D.; Ki, Z.; Litchy, M.: Aerosol dynamics in the equatorial Pacific marine boundary layer: Microphysics, diurnal cycles and entrainment, Geophys. Res. Lett. 23, 733-736, doi: https://doi.org/10.1029/96GL00778, 1996.

Quinn, P., Coffman, D., Johnson, J., Upchurch, L. M., Bates, T. S.:Small fraction of marine cloud condensation nuclei made up of sea spray aerosol. *Nature Geosci, 2017*, 10, 674–679, doi: https://doi.org/10.1038/ngeo3003, 2017.

**Vertical profile of smoke transport off of Africa.**

Anderson, B. E., Grant, W. B., Gregory, G. L., Browell, E. V., Collins Jr., J. E., Sachse, G. W., Bagwell, D. R., Hudgins, C. H., Blake, D. R., and N. J. Blake, N. J., (1996), Aerosols from biomass burning over the tropical South Atlantic region: Distributions and impacts, *J. Geophys. Res.*, 101(D19), 24117–24137, doi:10.1029/96JD00717.

Dobracki, A., Lewis, E. R., Sedlacek III, A. J., Tatro, T., Zawadowicz, M. A., and Zuidema, P.: Burning conditions and transportation pathways determine biomass-burning aerosol properties in the Ascension Island marine boundary layer, Atmos. Chem. Phys., 25, 2333–2363, https://doi.org/10.5194/acp-25-2333-2025, 2025.

Haywood, J. M., Abel, S. J., Barrett, P. A., Bellouin, N., Blyth, A., Bower, K. N., Brooks, M., Carslaw, K., Che, H., Coe, H., Cotterell, M. I., Crawford, I., Cui, Z., Davies, N., Dingley, B., Field, P., Formenti, P., Gordon, H., de Graaf, M., Herbert, R., Johnson, B., Jones, A. C., Langridge, J. M., Malavelle, F., Partridge, D. G., Peers, F., Redemann, J., Stier, P., Szpek, K., Taylor, J. W., Watson-Parris, D., Wood, R., Wu, H., and Zuidema, P.: The CLoud–Aerosol–Radiation Interaction and Forcing: Year 2017 (CLARIFY-2017) measurement campaign, Atmos. Chem. Phys., 21, 1049–1084, https://doi.org/10.5194/acp-21-1049-2021, 2021.

Mallet, M., Nabat, P., Zuidema, P., Redemann, J., Sayer, A. M., Stengel, M., Schmidt, S., Cochrane, S., Burton, S., Ferrare, R., Meyer, K., Saide, P., Jethva, H., Torres, O., Wood, R., Saint Martin, D., Roehrig, R., Hsu, C., and Formenti, P.: Simulation of the transport, vertical distribution, optical properties and radiative impact of smoke aerosols with the ALADIN

regional climate model during the ORACLES-2016 and LASIC experiments, Atmos. Chem. Phys., 19, 4963–4990, https://doi.org/10.5194/acp-19-4963-2019, 2019.

Swap, R. J., Annegarn, H. J., Suttles, J. T., Haywood, J., Helmlinger, M. C., Hely, C., Hobbs, P. V., Holben, B. N., Ji, J., King, M. D., Landmann, T., Maenhaut, W., Otter, L., Pak, B., Piketh, S. J., Platnick, S., Privette, J., Roy, D., Thompson, A. M., Ward, D., and Yokelson, R.: The Southern African Regional Science Initiative (SAFARI 2000): overview of the dry season field campaign, South African J. Sci., 98, 125–130, 2002.

**CAMP2Ex papers on detrainment**

Hilario, M. R. A., and Coauthors, 2025: Quantifying Scavenging Efficiencies of Different Aerosol Species and Size-Resolved Volume Concentrations in Tropical Convective Clouds over the West Pacific. *J. Atmos. Sci.*, **82**, 267–282, <a href="https://doi.org/10.1175/JAS-D-24-0064.1">https://doi.org/10.1175/JAS-D-24-0064.1</a>.

Xiao, Q., Zhang, J., Wang, Y., Ziemba, L. D., Crosbie, E., Winstead, E. L., Robinson, C. E., DiGangi, J. P., Diskin, G. S., Reid, J. S., Schmidt, K. S., Sorooshian, A., Hilario, M. R. A., Woods, S., Lawson, P., Stamnes, S. A., and Wang, J.: New particle formation in the tropical free troposphere during CAMP2Ex: statistics and impact of emission sources, convective activity, and synoptic conditions, Atmos. Chem. Phys., 23, 9853–9871, https://doi.org/10.5194/acp-23-9853-2023, 2023.

---

## Author Comment (AC2)

This paper documents aerosol data and observations from the 2019 ONR PISTON cruise and NASA CAMP2Ex flights performed over the north-western part of the Maritime continent on South-East Asia. The manuscript is rather large and highly descriptive of the multiple aerosol processes going on the Northwest Tropical Pacific's monsoon environment when multiple meteorological conditions are present during the observation period. The manuscript does an excellent job in advancing our current understanding of the interactions between meteorology, aerosols, and clouds on clean and polluted environments from a double perspective provided by aircraft and cruise measurements. This publication indeed does move the research frontier forward by providing a highly detailed description not only of the data itself but of the physics behind these multiple processes observed. Vertical and long-range transport of aerosol species and its presence over long distances and periods of time has been one of the areas for which very little measurements exist or none, let alone knowledge of its impact and implications for the *Maritime continent regional environment. In addition, sections 4.4 and 5.2.4 relating the role of* cold pools to particle lofting is, in my view, the biggest take home message and a very significant contribution to our understanding of aerosol lofting across this region. In essence, this work provides a fundamental baseline for future observations over the maritime region. I whole hearty recommend this article to be published and I would suggest to the authors to make a full review of the paper and identify areas to be condensed if possible.

Thank you for the positive review and for taking the time to thoroughly evaluate the paper. The team dedicated a great deal of effort to this work. As with Reviewer #1, we acknowledge that this paper is non-traditional in its length and scope, and thus presented a challenging review task. At the point of submission, we had already condensed the material considerably. Even certain elements that could potentially be relegated to a supplement, such as the comparison of airborne and vessel-based HSRL during a period of quiescence, were deemed fundamental to the paper and therefore retained in the methods section. Likewise, the mean profiles, while perhaps not central to the main thrust of the paper, also seemed out of place in the supplementary materials. We welcome any specific suggestions you may have regarding areas for further consolidation.

A couple of issues to note as shown below.

Line 550: "We surmise that these features..." I would guess the authors means "We assume ..." or something along those lines.

Thanks for the comment. Surmise is indeed the word that is appropriate here. Assume means that you are taking without evidence as a start of a proof. Surmise means essentially an educated guess, or to hypothesize. We switched the wording to hypotheses to add clarity.

Line 564: Table 2 is missing from the manuscript (there are other places in which Table 2 is mentioned).

Sorry, about that, it was missed when we entered the field into the template.

Line 590: "We initially surmised that the smoke..." Same typo as in Line 550 above.

Changed to hypothesized.

Line 940: Closing parenthesis is missing at the end of the paragraph.

Parentheses fixed